# The evolutionary advantage of an aromatic clamp in plant family 3 glycoside exo-hydrolases

Sukanya Luang[1,10], Xavier Fernández-Luengo[2,10], Alba Nin-Hill [3,10], Victor A. Streltsov [4], Julian G. Schwerdt[1], Santiago Alonso-Gil[3], James R. Ketudat Cairns [5], Stéphanie Pradeau [6], Sébastien Fort [6], Jean-Didier Maréchal [2], Laura Masgrau [2,7], Carme Rovira [3,8] & Maria Hrmova [1,9] ✉

In the barley β-ᴅ-glucan glucohydrolase, a glycoside hydrolase family 3 (GH3) enzyme, the Trp286/Trp434 clamp ensures β-ᴅ-glucosides binding, which is fundamental for substrate hydrolysis during plant growth and development. We employ mutagenesis, high-resolution X-ray crystallography, and multi-scale molecular modelling methods to examine the binding and conformational behaviour of isomeric β-ᴅ-glucosides during substrate-product assisted processive catalysis that operates in GH3 hydrolases. Enzyme kinetics reveals that the W434H mutant retains broad specificity, while W434A behaves as a strict (1,3)-β-ᴅ-glucosidase. Investigations of reactant movements on the nanoscale reveal that processivity is sensitive to mutation-specific alterations of the tryptophan clamp. While wild-type and W434H utilise a lateral cavity for glucose displacement and sliding of (1,3)-linked hydrolytic products through the catalytic site without dissociation, consistent with their high hydrolytic rates, W434A does not adopt processive catalysis. Phylogenomic analyses of GH3 hydrolases disclose the evolutionary advantage of the tryptophan clamp that confers broad specificity, high catalytic efficiency, and processivity.

The barley (*Hordeum vulgare* L.) β-ᴅ-glucan glucohydrolase HvExoI is the only plant family 3 glycoside hydrolase (GH3) for which the atomic structure is available[1–3], although the Carbohydrate-Active Enzymes CAZy database[4] lists nearly 48,400 GH3 entries (August 2022) from archaea, bacteria, fungi, and eukaryotes, including economically important plants *Triticum*, *Oryza*, *Glycine*, *Zea*, *Brassica*, *Medicago*, *Phaseolus*, and *Solanum*.

HvExoI folds into a two-domain structure consisting of an (α/β)₈ barrel (domain 1) and an (α/β)₆ sandwich (domain 2) with a 13 Å-deep active site pocket embracing the subsites −1 and +1, positioned at the interface between the two domains. At the −1 subsite, the pocket contains the catalytic nucleophile Asp285 and the acid/base catalyst Glu491, while the +1 subsite is shaped in a clamp with a coin slot formed by the Trp286 and Trp434 residues that hold β-ᴅ-glucosides entering the active site in a productive mode for hydrolysis[1–3,5] (Fig. 1a, b). HvExoI catalyses the hydrolytic removal of nonreducing glucose (Glc) moiety from a broad spectrum of poly- and oligomeric β-ᴅ-glucosides. This hydrolytic reaction underpins a vital role of HvExoI during plant growth and development, to participate during *e.g.* primary root extension and the recycling of mixed-linkage (1,4;1,3)-β-ᴅ-glucans during germination and root elongation[1,3,6]. The capacity of HvExoI to hydrolyse a panel of β-ᴅ-glucosides was rationalised from the observation that non-reducing Glc at the −1 subsite is locked into a

**Fig. 1 | Exo-hydrolytic enzymes of the GH3 family with an entrapped Glc molecule and a ligand-free form, and their phylogeny. a** *Hordeum vulgare* HvExoI with entrapped Glc (PDB 3WLH; atomic colours), *Bacteroides ovatus* β-glucosidase with perfused Glc (PDB 5JP0; green cpk), and a ligand-free form of *Pseudoalteromonas* sp. exo-(1,4;1,3)-β-glucanase (PDB 3UT0; yellow cpk). Distances ached of HvExoI residues to Glc in the −1 subsite (black dashes) are within 3.3 Å. HvExoI and structurally equivalent residues in 3UT0 and 5JP0 superpose with respective RMSD values of 0.7 Å and 1.2 Å. **b** Surface morphology of the active site pocket of HvExoI coloured by an electrostatic potential: white, neutral; red, −5 kT·e⁻¹. Separations (dashed lines) between Glc and key interacting residues are within 3.3 Å. The positions of the −1 subsites in panels **a** and **b** are indicated. **c** Unrooted phylogenetic tree of representative GH3 family members (accession numbers listed in Supplementary Table 1) revealed the presence of four major clades with five activities [β-D-glucan glucohydrolases (EC 3.2.1.-), β-D-glucosidases (EC 3.2.1.21), β-N-acetylhexosaminidases (EC 3.2.1.52), β-D-xylosidases (EC 3.2.1.37), and α-L-arabinofuranosidases (EC 3.2.1.55)]. In β-D-glucan glucohydrolases, three subclades were identified with residue variability in the aromatic clamp (residue numbering corresponding to HvExoI; PDB 1EX1). The percentage of replicate trees with taxa clustering together in the bootstrap test (1000 replicates) are shown at node bipartitions.

fixed position through a network of hydrogen bonds (H-bonds), while the reducing Glc moiety at the +1 subsite is constrained through stacking interactions, and thus the flexibility of attached substrates and their projection is unencumbered allowing HvExoI to accommodate isomeric (1,2)-, (1,3)-, (1,4)- and (1,6)-linked β-D-glucosides[2,3]. It was estimated that the strength of binding through aromatic residues could be in the range of 2-5 kcal/mol[7].

Since the HvExoI structure was elucidated[1–3], many prokaryotic, fungal and unclassified metagenome GH3 structures have become available but provided little information on the substrate specificity of

this class of enzymes. These include two-domain bacterial β-N-acetylglucosaminidases, for example, from *Bacillus subtilis*[8], and a β-N-acetylhexosaminidase from *Salmonella typhimurium*[9], while a β-hexosaminidase from *Burkholderia cenocepacia*[10] folds into a single-domain $(\alpha/\beta)_8$ sandwich structure. Several three-domain architectures consisting of an $(\alpha/\beta)_8$ barrel, an $(\alpha/\beta)_6$ sandwich, and another β-sandwich domain were revealed in β-glucosidases from *Thermotoga neapolitana*[11], *Hypocrea jecorina*[12] and *Aspergillus aculeatus*[13], in a β-N-acetylglucosaminidase from *Thermotoga neapolitana*[14], in a (1,4;1,3)-β-exo-glucanase from *Pseudoalteromonas* sp. BB1[15], and in a β-glucosidase JMB19063 isolated from metagenome[16]. These additional β-sandwich domains are described as fibronectin type III or Ig-like β-sandwich folds and have similar architectures. Four-domains structures consisting of $(\alpha/\beta)_8$ barrel, an $(\alpha/\beta)_6$ sandwich, and two β-sandwich domain folds (PA14 and fibronectin type III) were identified in the β-glucosidases from *Kluyveromyces marxianus*[17], *Aspergillus fumigatus*[18], *A. oryzae*[18] and *Streptomyces venezuelae*[19]. The precise roles of these additional β-sandwich domains have not been fully clarified. However, in some GH3 enzymes these domains are thought to be linked to substrate binding and specificity[17], protein structure stability[14,20], and oligomerisation, meaning that dimerisation is required for catalytic activity[16].

Although the details of the catalytic mechanism in HvExoI were reported[1–3,21], the precise information on the binding of positionally isomeric β-D-glucosides at the atomic level remains limited. Recently, we envisaged that β-D-glucan glucohydrolases, such as HvExoI, adopt so-called substrate-product assisted processive catalysis, where after an incoming substrate binds in a productive mode, the Glc product evokes the formation of a transient lateral cavity, which serves as a conduit for its egress to allow for the next catalytic cycle[22]. To our knowledge, this type of processive catalysis (or non-dissociative sequential degradation) discovered in HvExoI[22] has not been described in hydrolases with a closed pocket-shaped catalytic site. In doing so[22], we defined processive catalysis (which we termed substrate-product assisted processivity) in broader terms than those used for the definitions of the tunnel-, groove-, ridge-, or cleft-like active sites of GHs[23–25]. Considering that HvExoI operates on plant cell wall (1,3;1,4)-β-D-glucan polysaccharides, processive catalysis (as a form of efficient catalysis) in this enzyme is expected as rapid substrate hydrolysis is critical to embryo development during seed germination[5,26]. Processive catalysis has been defined in (1,3)-β-D-glucan endohydrolases[27], (1,4)-β-D-glucan endohydrolases[28], cellobiohydrolases[23], and chitobiohydrolases[24,25,29]. Conversely, exo-acting hydrolases[21,22] and β-D-glucosidases[30] with closed funnel-, crater-, or pocket-shaped active sites have been deemed to be nonprocessive[30]. The work with processive enzymes suggested the role of aromatic and polar amino acid residues to assist with stacking and H-bond interactions for polysaccharide sliding motions through the catalytic sites[22–25,27–29,31].

In this context, our aim was to provide direct structural evidence for the binding of β-D-glucosides and the unbinding of their hydrolytic products and investigate the relationship between reactant movements and processivity.

Here, we capitalise on a unique feature of HvExoI to bind a panel of isomeric β-D-glucosides through the aromatic clamp of two Trp residues, and elucidate, how this precisely facilitates substrate binding, catalytic efficiency, and substrate-product assisted processivity. We present atomic details of HvExoI wild-type (WT) and its Ala, His, Phe, and Tyr mutants of the Trp286/Trp434 clamp in the complex with positionally isomeric thio-analogues. Based on these structures, we evaluate the reactant movements on the nanoscale in HvExoI via classical molecular dynamics (cMD), Gaussian accelerated MD (GaMD) simulations and GPathFinder calculations. These findings combined with modelling of the conformational behaviour of reactants via cMD and quantum mechanics/molecular mechanics (QM/MM) MD

metadynamics, indicate that in this GH3 exo-hydrolase the aromatic clamp fulfils the role of a recognition platform ensuring effective carbohydrate-binding due to cooperative rotational and translational movements. This knowledge expands on the applicability of bioengineering of carbohydrate-processing enzymes to improve catalytic rates, and stability, and to understand product inhibition (important in drug discovery), as the traditional industrial approaches are being increasingly replaced by enzyme-based technologies.

## Results

### Mutants altered in Trp286 and Trp434 residues

The productive binding of carbohydrates in HvExoI involves the participation of Trp286 and Trp434 residues that form an aromatic clamp at the +1-subsite, while the neighbouring subsite contains the Asp285 and Glu491 catalysts[1–3] (Fig. 1a). In HvExoI, after the hydrolytic cycle is completed, both subsites contain entrapped Glc products that are displaced after a new substrate is attached[22]. The Glc molecule entrapped in the −1 subsite (Fig. 1b) is held at the bottom of the pocket by at least 10 residues (Fig. 1a, b). The residues equivalent to the HvExoI 286 and 434 positions overlap in the homologous GH3 exo-hydrolases[14,15,32], form similar aromatic clamps, and that of *Bacteroides* β-glucosidase (PDB 5JP0) harbours the Glc molecule after *in crystallo* perfusion (Fig. 1a). To resolve phylogenetic relationships of HvExoI within the GH3 family[4] a phylogenetic tree of selected entries was built revealing that the entries segregated in four clusters harbouring five enzyme activities: β-D-glucan glucohydrolases, β-D-glucosidases, β-N-acetylhexosaminidases, β-D-xylosidases, and α-L-arabinofuranosidases[33,34] (Fig. 1c; Supplementary Table 1). The inspection of the β-D-glucan glucohydrolase cluster revealed that Trp286 and Trp434 remained conserved in subclade 1, while in the subclades 2 and 3 enzymes, we identified variations in both Trp residues – Trp286/Tyr286 and Gln434/Tyr434/Trp434 in subclade 2, and Gly434/His434/Lys434/Trp434 in subclade 3 (Fig. 1c), which may impact substrate binding and processivity in these enzymes.

To examine the precise mechanistic roles of the Trp286 and Trp434 residues in the binding of isomeric (1,2)-, (1,3)-, (1,4)- and (1,6)-linked β-D-glucosides in HvExoI, we methodically varied these residues to Ala, His, Phe, and Tyr to generate single (and in some instances) double mutants, and investigated their substrate specificity, steady-state and inhibitions kinetics and substrate binding in atomic structures (Figs. 2–4; Supplementary Figs. 2–4; Supplementary Tables 2–6).

### Substrate specificity of WT, and Trp286 and Trp434 mutants

We compared relative hydrolytic rates of β-linked poly- and oligo-saccharides, and aryl-glycosides under optimal physico-chemical conditions (Supplementary Fig. 1; Supplementary Table 2; Supplementary Note 1). Of the tested mutants, W434H was the only mutant that exhibited broad substrate specificity similar to WT, contrary to W286F and W286Y, and W434F, W434Y, W286F/W434F, and W286F/W434A that lacked activities of at least with one carbohydrate substrate (Supplementary Table 2). The absence of hydrolytic activities was observed with (1,4;1,3)-β-D-glucans (barley glucan and lichenin) and 4NP-cellobioside in all mutants except W434H and a decrease in hydrolytic rates with oligomeric substrates and aryl β-D-glucosides except in W286Y with 4-nitrophenyl β-D-glucopyranoside (4NP-Glc) (Supplementary Table 2). Remarkably, W434A lost all hydrolytic activities with poly- and oligosaccharides, except (1,3)-β-D-linked β-D-glucosides, turning this mutant into a strict (1,3)-β-D-glucosidase, albeit with low specific activities (Supplementary Table 2). Our findings suggest that although the mutations of Trp286 to Ala and His led to folded but inactive enzymes, enzyme activity retention is more sensitive to alterations of Trp286 than Trp434, likely due to the proximity of Trp286 to Asp285, the catalytic nucleophile (Supplementary Table 2).

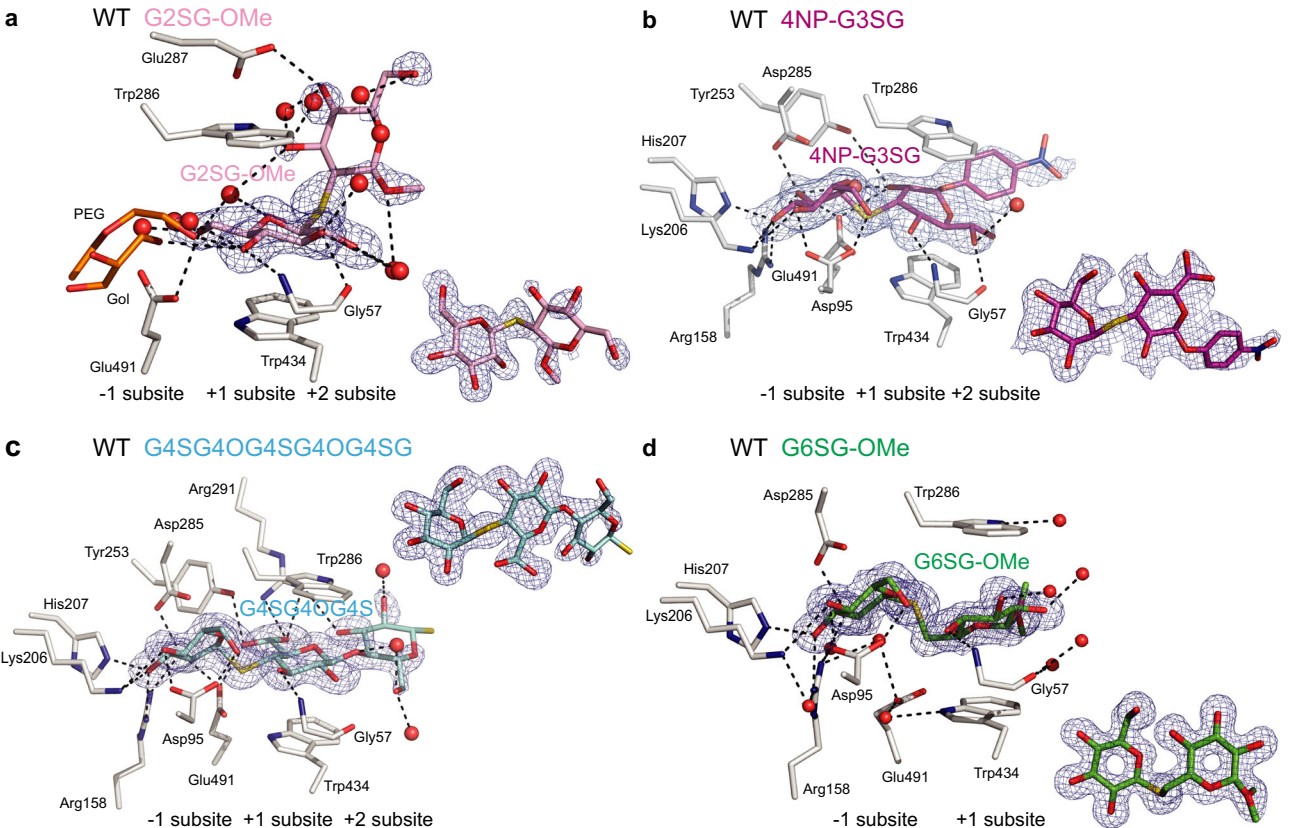

**Fig. 2 | WT HvExoI in complex with thio-saccharide analogues. a** Thio-sophoroside (G2SG-OMe; pink); **b** thio-laminaribioside (4NP-G3SG; magenta); **c** thio-cellotrioside (G4SG4OGS; cyan); and **d** thio-gentiobioside (G6SG-OMe; green) moieties are bound across the −1 to +2 subsites. Panels **a** to **d** contain derived difference 2 m|$F_o$|-D|$F_c$| electron density maps (blue) of thio-analogues (also shown as insets rotated by around −90° via x-axes relative to main panels) contoured at 1.0 σ levels. Glycerol (Gol) or polyethylene glycol (PEG) and water molecules are shown in cpk orange sticks and red spheres, respectively. Distances at separations within 3.3 Å are shown in dashed lines. Ligand designations above structural images indicate thio-analogues that were perfused in crystals.

## Steady-state kinetic constants of WT, and Trp286 and Trp434 mutants

Consistent with our predictions, catalytic rate $k_{cat}$ (first-order rate) constants and in some instances, Michaelis-Menten $K_M$ values of single W286A, W286H, W286F, W434A, W434H, W434F, and W434Y, and double W286F/W434F and W286F/W434A mutants with poly- and oligomeric substrates decreased compared to WT (Supplementary Table 3). This trend continued with the catalytic efficiency $k_{cat}/K_M$ (second-order rate) constants, where these values corresponded to those of relative hydrolytic rates (Supplementary Table 2). These values confirmed the strict (1,3)-β-oligosaccharide specificity of W434A, with little change in the apparent strength of binding of laminaribiose, but by an about 21-fold decrease of $k_{cat}$ (compared to WT), consequently yielding a low $k_{cat}/K_M$ value. A similar profile was noted for W286F/W434A, although in this case, the kinetic parameters decreased even more, pointing to a negative effect of the non-conservative (radical) W434A mutation on substrate binding and hydrolysis (Supplementary Table 3). Notable was the behaviour of W286Y that showed increased catalytic efficiency values with (1,3;1,6)-β-D-glucan (laminarin), laminaribiose (G3OG), and 4NP-Glc, while these values in W434Y were considerably lower (Supplementary Table 3). In W286F/W434F, the catalytic efficiency values were also drastically reduced compared to WT, whereas the W434H and W434F mutants showed higher catalytic efficiency than the double mutants, with W434F being more active except with laminaribiose. These data signal that although the mutations of Trp434 affected the catalytic properties of HvExoI more substantially compared to Trp286, both Trp residues are required for binding the isomeric β-D-glucosides to achieve productive binding modes required for high catalytic efficiency.

## Inhibition constants $K_i$ of WT, and Trp286 and Trp434 mutants

To further examine binding modes of β-D-glucosides, we used the thio-analogues with (1,2)-, (1,3)-, (1,4)- and (1,6)-linkages to quantify apparent inhibition constants ($K_i$) of the HvExoI:inhibitor complexes (Supplementary Table 4). In WT HvExoI, the (1,3)-β-D-linked (4NP-G3SG), (1,4)-β-D-linked (G4SG-OMe), and (1,6)-β-D-linked (G6SG-OMe) thio-analogues showed a competitive inhibition with $K_i$ values in low to medium micromolar ranges, except G2SG-OMe, which was weakly inhibitory (Supplementary Table 4). The W434A mutant was sensitive to 4NP-G3SG and G4SG-OMe, and the data with 4NP-G3SG correlated with substrate specificity and catalytic efficiency (Supplementary Tables 2 and 3). As expected, the (1,3)- and (1,4)-linked thio-inhibitors were less potent with tested mutants, while the most effective thio-inhibitor was G6SG-OMe, except that it did not bind to W434A, and compared to WT, it showed 8- and 6-fold higher $K_i$ constant values with W434H and W434F, respectively (Supplementary Table 4). These data pointed out that the replacements of Trp434 with Ala, His, or Phe residues in the +1 subsite, disturbed the optimal binding of thio-analogues, due to the removal or weakening stacking interactions crucial for the correct positioning of inhibitors in the active site.

## WT and mutant HvExoI:thio-β-D-glucoside complexes reveal the details of binding at the atomic level

We observed that the thio-analogue $K_i$ constants (except G2SG-OMe) measured with the Trp286 and Trp434 mutants, correlated with their capacity to hydrolyse (1,3)-, (1,4)- and (1,6)-linked β-D-glucosides. To decipher their precise binding modes at the atomic levels, we pursued structural studies (Figs. 2–4; Supplementary Figs. 2–4; Supplementary Tables 5 and 6).

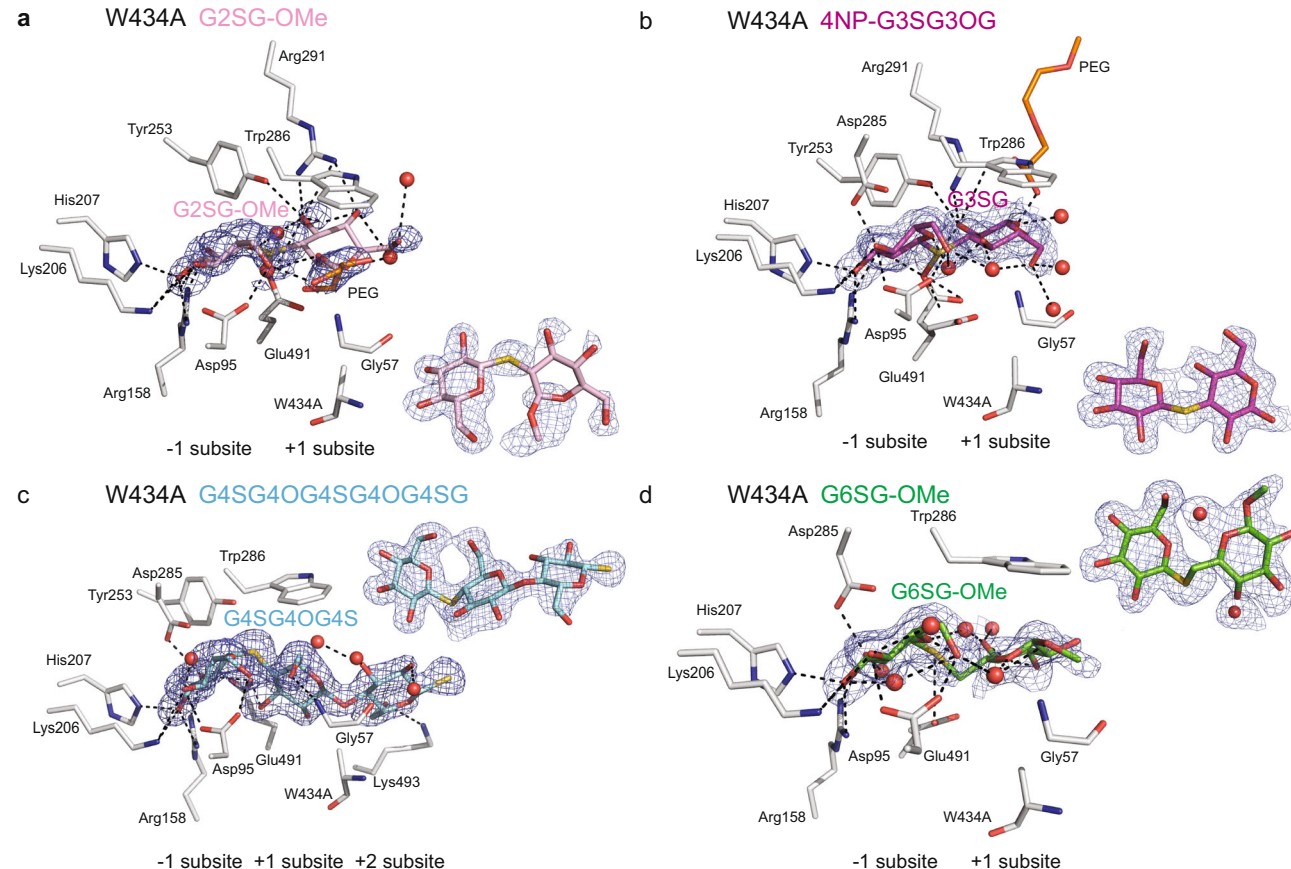

**Fig. 3 | Thio-saccharide analogues are bound in the active site of the HvExoI W434A mutant. a** Thio-sophoroside (G2SG-OMe, pink); **b** thio-laminaribioside (4NP-G3SG, magenta); **c** thio-cellotrioside (G4SG4OGS; cyan); and **d** thio-gentiobioside (G6SG-OMe, green) moieties are bound across the −1 to +2 subsites. Panels **a**–**d** contain derived difference $2\,m|F_o|\text{-D}|F_c|$ electron density maps (blue) of thio-analogues (also shown as insets rotated by around −90° via x-axes relative to main panels) contoured at 1.0 σ levels. Polyethylene glycol (PEG) and water molecules are shown in cpk orange sticks and red spheres, respectively. Distances at separations within 3.3 Å are shown in dashed lines. Ligand designations above structural images indicate thio-analogues that were perfused in crystals.

**G2SG-OMe binding.** The G2SG-OMe thio-analogue bound weakly to native and WT recombinant HvExoI with the respective $K_i$ of 2 $\times 10^{-3}\,M$ and 4 $\times 10^{-3}\,M$ (Supplementary Table 4), reflecting its ability to attach at the +1 and +2 subsites instead of the −1 and +1 subsites (Fig. 2a)[22], which was most likely due to the rigidity of this thio-ligand. On the other hand, in W286F the electron density map of G2SG-OMe was defined at the −1 and +1 subsites (Supplementary Fig. 2c), where this thio-ligand formed H-bonds with Asp95, Lys206, His207, Arg158, Tyr253, Asp285, Arg291, and Glu491, and stacking interactions with W286F and Trp434, whereas G2SG-OMe did not bind to W286A and W286Y. The dispositions of G2SG-OMe at the −1 and +1 subsites in the W434A, W434F, W434H, and W434Y mutants (Figs. 3a, 4a; Supplementary Figs. 3a and 4a) revealed similar structural features of the ligand binding to those seen in W286F (Supplementary Fig. 2c).

**4NP-G3SG and 4NP-G3SG3OG binding.** 4NP-G3SG bound tightly to native and WT recombinant HvExoI with the $K_i$ constant of 0.7 $\times 10^{-3}\,M$ (Supplementary Table 4); the $K_i$ value of 0.2 $\times 10^{-3}\,M$ was previously observed with 4NP-G3SG3OG in WT HvExoI[3]. These relatively tight binding constants suggested that these thio-analogues should be seen in the active sites, as we observed with WT, and the W286F, W434A, W434F, W434H, and W434Y mutants. The well-defined electron density maps of the (1,3)-linked thio-ligands illustrated that the saccharide moieties were attached through the networks of H-bonds formed between nine to ten residues in each case (Figs. 2b, 3d, 4b; Supplementary Figs. 2d, 3b, 4b). The 4NP-G3SG thio-analogue in WT was

located at the +1 to +2 subsites, with the 4NP group clearly discernible at the +2 subsite (Fig. 2b). However, in W286A, and W286Y, 4NP-G3SG was not observed, and in W286F, the reducing-end Glc moiety of 4NP-G3SG at the +1 subsite remained disordered (Supplementary Fig. 2d). This emphasised that the Trp286 mutations to Ala, Tyr, and Phe disturbed the binding of (1,3)-linked thio-β-D-glucosides and that this residue was necessary for inhibitor binding.

**G4SG-OMe and G4SG4OG4SG4OG4SG binding.** G4SG-OMe bound to native and WT recombinant HvExoI with the $K_i$ values of 1.0 $\times 10^{-3}\,M$ (Supplementary Table 4). This compares favourably to the hexasaccharide G4SG4OG4SG4OG4SG, where we previously observed the $K_i$ constant of 0.6 $\times 10^{-3}\,M$ with native HvExoI[2]. These binding constants suggested that the (1,4)-linked thio-analogues should locate in the enzyme's active sites. This occurred with WT, and the W434A, W434F, W434H, and W434Y mutants, while no binding was observed in the W286A, W284F, and W286Y mutants. The well-defined electron density maps in W434A (Fig. 3c) and W434F (Supplementary Fig. 3c) revealed three saccharide moieties at the −1 to +2 subsites with the reducing thio-saccharide moiety protruding beyond the environment of the active site pocket at the +2 subsite. In W434A, we observed that the tri-thio-cellobioside moiety was somewhat misplaced (compared to WT) and that the W434A substitution unexpectedly allowed Lys493 to be rotated outward and become engaged in the binding of this thio-ligand; here, 11 residues participated in forming H-bonds with tri-thio-cellobioside moiety (Fig. 3c). To generate the W434H and W434Y complexes, we perfused the G4SG-OMe ligand, consequently, only two

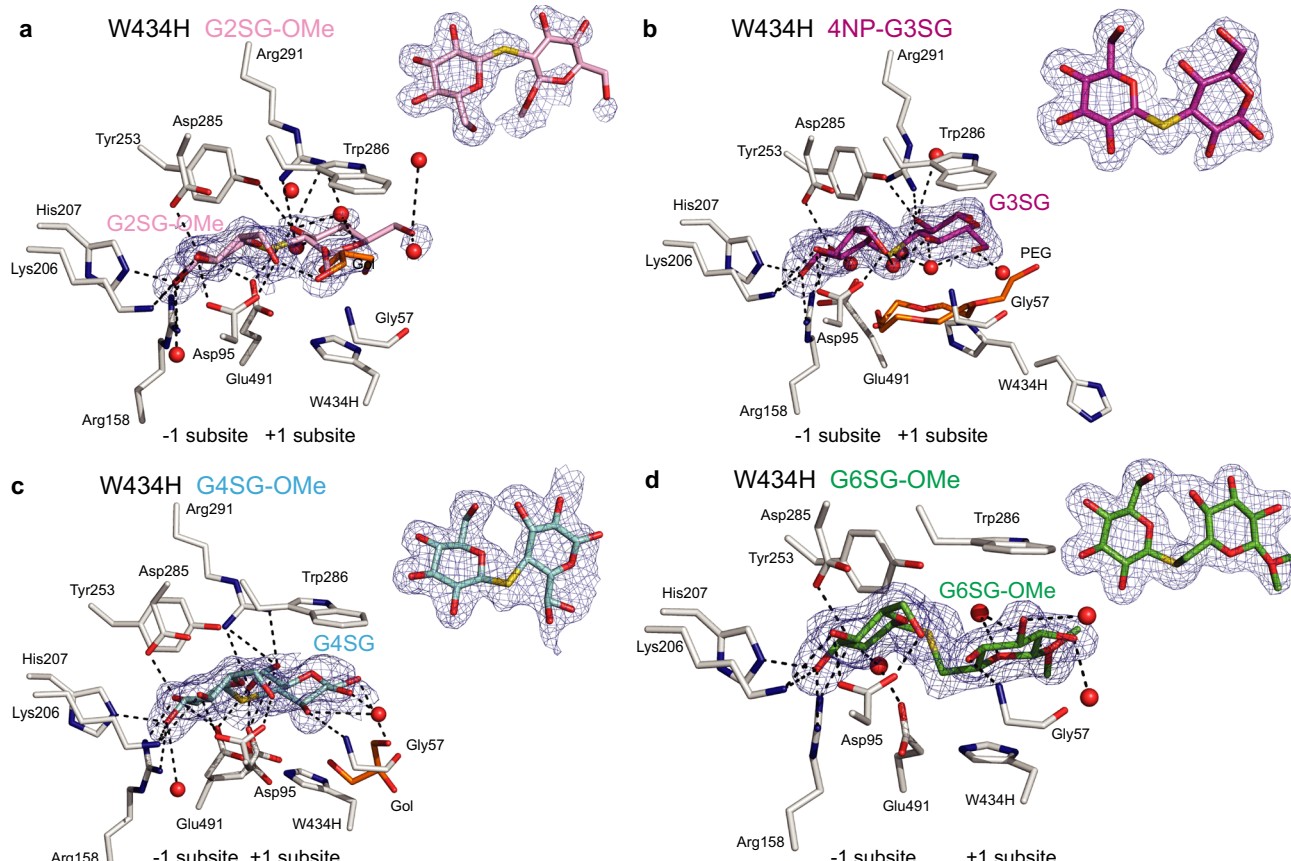

**Fig. 4 | Thio-saccharide analogues are bound in the active site of the HvExoI W434H mutant. a** Thio-sophoroside (G2SG-OMe, pink); **b** thio-laminaribiose (4NP-G3SG, magenta); **c** thio-cellobiose (G4SG; cyan); and **d** thio-gentiobiose (G6SG-OMe, green) moieties are bound across the −1 to +2 subsites. Panels **a** to **d** contain derived difference 2 m$|F_o|$-D$|F_c|$ electron density maps (blue) of thio- analogues (also shown as insets rotated by around −90° via x-axes relative to main panels) contoured at 1.0 σ levels. Polyethylene glycol (PEG) and water molecules are shown in cpk orange sticks and red spheres, respectively. Distances at separations within 3.3 Å are shown in dashed lines. Ligand designations above structural images indicate thio-analogues that were perfused in crystals.

thio-saccharide moieties were observed in active sites (Fig. 4c; Supplementary Fig. 4c). But with W286H, and W286Y, we could not see the binding of G4SG-OMe, which correlated with their weaker $K_i$ values compared to WT (Supplementary Table 4), and suggested that the conservative Trp286 mutation to Phe and Tyr affected the binding of (1,4)-linked thio-β-D-glucosides.

**G6SG-OMe binding.** G6SG-OMe bound tightly to native and WT recombinant HvExoI with the respective $K_i$ values of 0.1 ×10$^{-3}$ M and 0.2 ×10$^{-3}$ M (Supplementary Table 4), hinting that this thio-analogue should be observed in the active sites of WT and mutants (Figs. 2d, 3d, 4d; Supplementary Figs. 2a, b, e, 3d, 4d). This is what we indeed observed in WT, and the W286A, W286Y, W286F, W434A, W434F, W434H, and W434Y mutants, where well-defined electron density maps of G6SG-OMe and tight H-bond networks that were formed through eight to ten residues (Asp95, Lys206, His207, Arg158, Tyr253, Asp285, Arg291, Glu491, Trp286, and Trp434). In W434A we noticed approximately 120° rotation (through the x-axis) of the reducing-end moiety of G6SG-OMe compared to its disposition in WT (or in-solution structure of G6OG or gentiobiose), due to the presence of a rotatable C1-O-C6 glycosidic bond of G6SG-OMe; this did not occur in WT or other mutants. To find out, if the Glc molecule alone adopts similar or different dispositions at the −1 and +1 subsites, we perfused Glc into WT, W434A, and W286Y, and compared their positions, which at the −1 subsite were identical in all instances, but at the +1 subsite of W434A the Glc molecule was disordered (data not shown). We noted that the quality of the G6SG-OMe electron density map slightly declined in

W286Y (Supplementary Fig. 2b) but that these maps were well-formed in W286A and W286F (Supplementary Fig. 2a, e), suggesting that in the W286Y mutant the process of ligand-binding in (or near) the active site may have been perturbed. These findings reinforce the conclusion that G6SG-OMe binds in HvExoI regardless of mutations (particularly into aromatic residues) of Trp286 or Trp434. Although these mutations impact the precise orientation of the glucopyranoside moieties, as observed in W434A, they permit G6SG-OMe binding.

**Conformational behaviour of S- and O-linked β-D-glucosides**
To determine the conformational landscape of β-D-glucosides bound in HvExoI, we analysed enzyme-thio-ligand-binding modes in the crystal structures before and after cMD simulations. As described above, the β-D-glucopyranose rings of thio-saccharide analogues at the −1 subsites in the WT crystal structure complexes were engaged in H-bond networks and observed in the $^4C_1$ conformation for G2SG-OMe and G6SG-OMe, the $^4H_3$ conformation for 4NP-G3SG, and the $^4E$/$^4H_3$ conformations for G4SG4OG (Supplementary Figs. 5 and 6; Supplementary Table 7), while the β-D-glucopyranose moieties of thio-ligands at the +1 subsites adopted the $^4C_1$ conformation. To analyse the dynamic behaviour of β-D-glucosides, a total of eight models were generated for G2(S or O)G, G3(S or O)G, G4(S or O)G, and G6(S or O)G systems, wherein we reverted mutations (if present) in HvExoI, removed ions and other molecules from thio-analogues, and for the O-linked systems, replaced S-atoms with O-atoms. While cMD simulations revealed that β-D-glucopyranose rings in thio-analogues adopted predominantly the $^4C_1$ conformations in both subsites, alternate

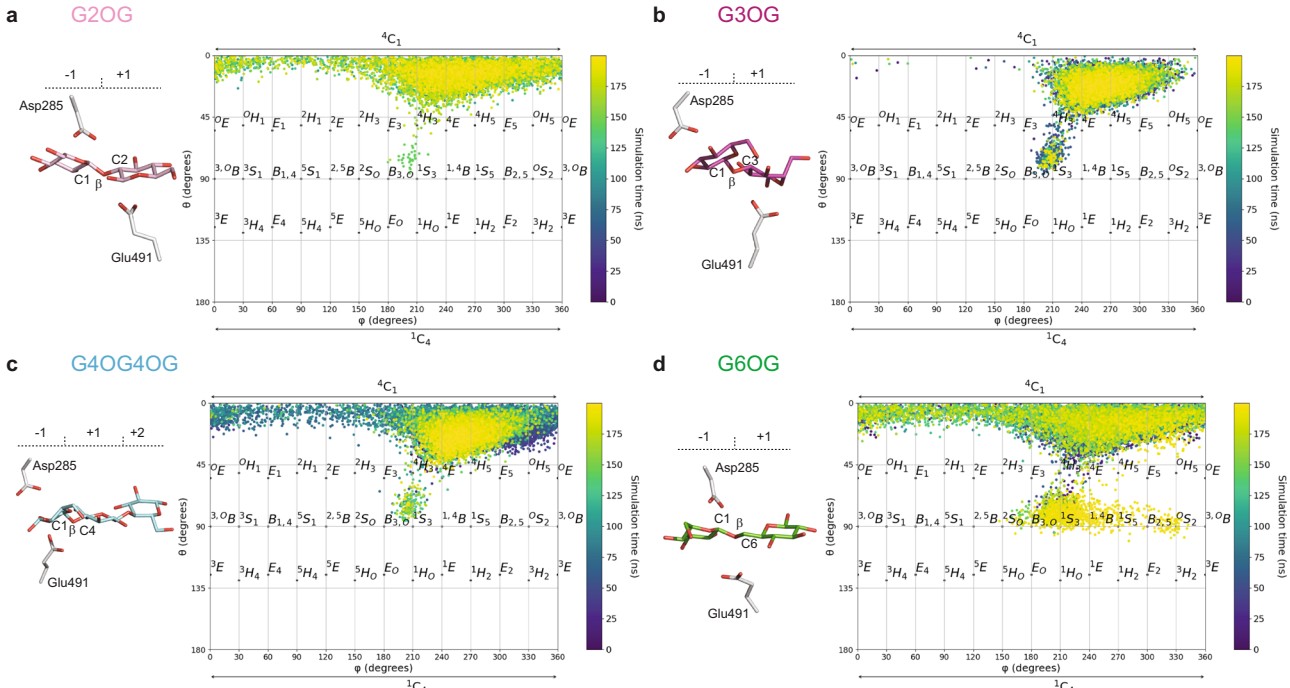

**Fig. 5 | Conformational behaviour of O-linked oligosaccharides bound in the active site of HvExoI calculated by cMD simulations plotted as a function of θ and φ puckering coordinates. a** Sophorose (G2OG; pink); **b** laminaribiose (G3OG; magenta); **c** cellotriose (G4OG4OG; cyan); and **d** gentiobiose (G6OG; green) ligands are bound at the −1 to +2 subsites (colour gradients from purple at 0 ns to yellow at 200 ns in Mercator projections are indicated). Complexes of WT with G3SG, G4SG4OG, G6SG, and W434A in complex with G2SG were used to predict binding modes of O-linked oligosaccharides. The β-D-glucopyranose moieties at the −1 to +2 subsites, and dispositions of Asp285 and Glu491 catalytic residues are also shown.

conformations also participated, such as $^4H_3/^4E/^4H_5$ or $^1S_3/^{1,4}B$, depending on a type of the glycosidic linkage (Fig. 5; Supplementary Fig. 6; Supplementary Table 7). Additionally, we reconstructed O-derivatives in the active sites based on the thio-analogue complexes and subjected these to cMD (Fig. 5; Supplementary Fig. 7; Supplementary Table 7). Compared to the analyses of the thio-analogue complexes discussed above, these data indicated that the β-D-glucopyranose rings of G2OG, G3OG, G4OG4OG, and G6OG adopted mainly $^4C_1$ conformations in both subsites, although other states contributed to binding in the −1 subsite (32% in the $^4H_5/^4E$ region for G3OG, and respective 5%, 10% and 27% in $^4H_3$, $^4E$, $^4H_5$ regions for G2OG, G6OG, and G4OG4OG). In the +1 subsite, the G3OG β-D-glucopyranose ring was located at respective 66% and 28% in the $^4C_1$ and $^2S_O$ regions, while the β-D-glucopyranose rings of G2OG, G6OG, and G4OG4OG occupied the $^4C_1$ regions in more than 90% of occurrences (Supplementary Table 7). Notably, the Mercator projection of G3OG (Supplementary Fig. 8) concurred with its conformational Free Energy Landscape (FEL) map (Supplementary Fig. 9), computed by QM/MM MD metadynamics. As recently found for other exo-acting GH enzymes[35,36], these findings predicted that HvExoI likely exhibits a two-fold hydrolytic itinerary with a positionally isomeric disaccharide such as G3OG, where the hydrolytic reaction starts either from a distorted $^4H_3/^4E/^4H_5$ region or from an undistorted $^4C_1$ β-D-glucopyranosyl ring, which would evolve towards a transition state with a $^4H_3/^4E$ conformation in both cases.

**Binding and hydrolysis of the G4OG3OG natural substrate were explored via cMD and QM/MM MD metadynamics simulations**
To understand the binding of one of the natural substrates of HvExoI, we selected the kinked G4OG3OG oligosaccharide (Fig. 6; Supplementary Figs. 10 and 11), which is the key hydrolytic product originating from (1,4;1,3)-β-D-glucans; these polymers fulfil the role of structural cell wall components in cereals, such as barley[5]. The complex with G4OG3OG was generated based on the WT HvExoI:G4SG4OG complex (Fig. 2c), upon replacing G4 with G3. The analysis of cMD

simulations of this complex showed that the Glc moieties of G4OG3OG adopted the stable $^4C_1$ conformations at the +1 and +2 subsites (Supplementary Fig. 10). However, Glc at the −1 subsite adopted a state, that was nearly identical to the Glc moiety in G3OG, and not entirely a perfect chair, thus we defined it as a pseudo-chair (Supplementary Fig. 10b). Again, this is consistent with the conformational FEL of G4OG3OG at −1 subsite, computed by QM/MM MD metadynamics (Fig. 6a; Supplementary Fig. 11), which shows a wide free energy minimum that extends from $^4C_1$ towards $^4H_3$.

Additionally, we investigated the binding of the G3OG product originating from the G4OG3OG substrate in WT and the W434A mutant, after the non-reducing Glc moiety was cleaved off, using cMD simulations (Figs. 6b, c). The cluster analysis of the trajectory of the G3OG product in WT indicated that the Glc moieties adopted at least two poses: ~96% of the populations located in the +1 and +2 subsites, and ~4% to the intermediary poses at the −1 and +1 subsites (Fig. 6b). Conversely, the cluster analysis of the G3OG product in W434A indicated a dramatic shift in poses to ~15% of the populations located in the +1 and +2 subsites, and ~74% to the intermediary (midway) poses at the −1 and +1 subsites (Fig. 6c). These observations imply that Trp434 plays a key role in the sliding of the (1,3)-β-D-linked hydrolytic products through the active site during catalysis and that Trp434 prevents the sliding of mixed-linkage substrates.

**Binding and displacements of laminaritriose (G3OG3OG) and laminarihexaose (G3OG3OG3OG3OG) hydrolytic products**
These studies were performed in the WT, W434H, and W434A enzymes with active sites accommodating hydrolytic products of laminaritriose, i.e. Glc in the −1 subsite and laminaribiose (G3OG) in the +1 and +2 subsites, or laminarihexaose, i.e. Glc at the −1 subsite and laminaripentaose (G3OG3OG3OG3OG) in the +1 and +2 subsites, and exposed to bulk solvent. The selection of these mutants was based on relative hydrolytic rates of poly- and oligosaccharide substrates, where these mutants either retain (W434H) or lose (W434A) their broad substrate

**a**   Conformational FEL map of Glc  at -1 subsite of G4OG3OG in WT

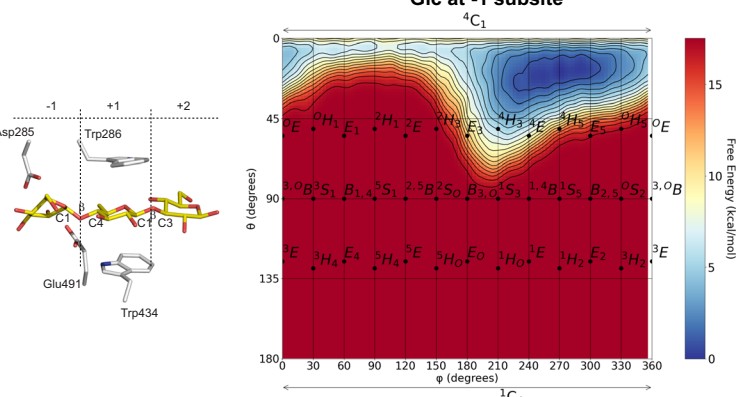

**b**   Behaviour of the G3OG product after hydrolysis from G4OG3OG in WT

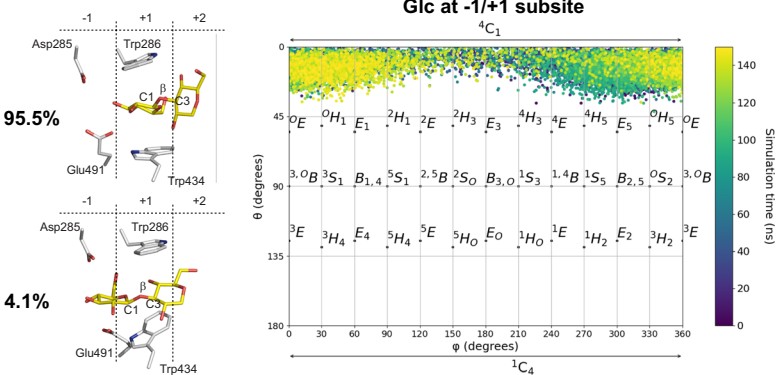

**c**   Behaviour of the G3OG product after hydrolysis from G4OG3OG in W434A

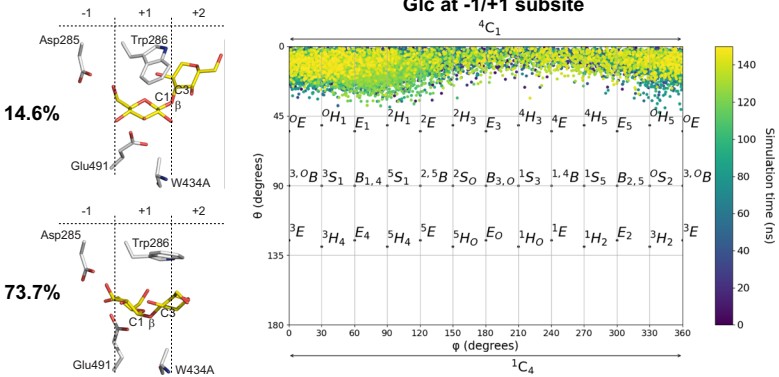

**Fig. 6 | Binding of the G4OG3OG natural substrate in the HvExoI active site calculated via cMD and QM/MM MD metadynamics simulations, plotted as a function of θ and φ puckering coordinates. a** Conformational FEL map of the Glc moiety at the −1 subsite in WT; Glc moieties of G4OG3OG in the subsites −1 to +2 are indicated along with the key residues; **b** conformational behaviour (Mercator projection) of the G3OG product after the −1 Glc moiety is hydrolysed from G4OG3OG and removed from the −1 subsite in WT; **c** conformational behaviour (Mercator projection) of the G3OG product after the −1 Glc moiety is hydrolysed from G4OG3OG and removed from the −1 subsite in W434A. Colour gradients from purple at 0 ns to yellow at 150 ns in Mercator projections are shown. In **b**, **c**, respective cluster analyses of trajectories (populations in %) indicate two modes of G3OG binding in WT and W434A, where each saccharide binds in the +1 and +2 subsites or Glc moieties bind in intermediary positions between those subsites and the −1 subsite.

specificity (Supplementary Table 2). To assess the effect of saccharides on HvExoI behaviour, cMD simulations were also performed with the WT apo-form, lacking bound Glc or β-ᴅ-glucosides.

The cMD and GaMD simulations with laminaritriose and laminarihexaose hydrolytic products indicated that the WT structure was stable (Supplementary Fig. 12), as indicated by backbone Root-Mean-Square-Deviation (RMSD) values, although high mobilities in some loops were detected. Conversely, cMD and GaMD simulations of W434H and W434A with laminaritriose

and laminarihexaose hydrolytic products showed contrasting behaviour compared to WT (Supplementary Fig. 12). cMD simulations of the mutants with the Glc and the G3OG products (run for 1000 ns) revealed that the protein backbones remained stable (respective RMSD values: 1.7 Å for W434H, and 1.25 Å for W434A) and that Glc was attached at the −1 subsite, but, after around 100 ns in W434H and 200 ns in W434A, G3OG moved to bulk solvent, indicating that the mutants were ineffective in retaining the G3OG product (Supplementary Fig. 13). Such behaviour was

not observed with laminarihexaose hydrolytic products, within the time-frames of cMD simulations (600 ns), whereby these products remained attached in WT and both mutants (Supplementary Fig. 13). These simulations indicate that the dynamics of ternary complexes of mutants presented variabilities compared to WT, where these differences depend on the length of hydrolytic products bound, i.e. longer products were more stable.

To resolve variability with the length of hydrolytic products in mutants, GaMD simulations (800 ns) with enhanced sampling were conducted. We found that the protein scaffold of the W434A mutant was more flexible (RMSD value 2.5 Å) than that of WT or W434H (RMSD values between 1.0 Å – 1.5 Å) (Supplementary Fig. 12), and that the Gly210-Met230 loop in the vicinity of the catalytic site exhibited Root-Mean-Square-Fluctuation (RMSF) values of up to 7 Å in both mutants compared to WT (RMSF value $\leq 2$ Å). While Glc remained highly restrained in the −1 subsite in WT and W434H, it was mobile in W434A, hinting that the overall flexibility of the catalytic site was little affected in W434H, but strongly affected in W434A. During GaMD simulations, the G3OG and laminaripentaose products remained bound in WT, but G3OG vacated the catalytic site in both mutants, in contrast to laminaripentaose, which initially remained bound (~25% and ~50% of simulation time for W434A and W434H, respectively), but eventually, it traversed to bulk solvent. These findings suggested that the replacements of Trp434 by His and specifically by Ala led to a loss of G3OG and laminaripentaose retention, and thus, a potential unbinding of these products could occur before the next catalytic cycle begins. This also implies that in W434H and W434A, but not in WT, the (1,3)-linked products would need to re-bind and cannot slide through the active site uninterrupted.

### Glc displacement route (egress) or substrate-product assisted processivity in WT

The modelling of the Glc displacement from the −1 subsite performed with GPathFinder[37], commenced from the most populated cluster of structures based on cMD, to further explore dynamic traits of Glc unbinding. These calculations were executed in two arrangements with (i) Glc at the −1 subsite and laminaripentaose in the +1 and +2 subsites and bulk solvent (Figs. 7, 8; Supplementary Movies 1–3), or Glc at the −1 subsite and G3OG in the +1 and +2 subsites (Supplementary Fig. 14); and (ii) Glc in the −1 subsite and the di- and pentasaccharide products removed to explore the Glc egress route without products being bound.

Initial simulations performed with WT (Glc at the −1 subsite and laminaripentaose in the +1 and +2 subsites and bulk solvent) yielded the Glc egress path (path 1) (Figs. 7a, 8; Supplementary Movie 1), which was facilitated by the re-orientation of a toll-like Arg158-Asp285-Glu491 barrier (including the side-chain of Asp285) and the motions of the loop carrying Tyr253. Next, Glc migration induced a rotation of Tyr253 that opened the lateral cavity due to the disruption of the Arg291-Glu220 salt bridge – these re-orientations and motions collectively opened the lateral cavity for Glc to exit. During these simulations, CH−π interactions between the non-reducing-end Glc moiety of laminaripentaose and the Trp286/Trp434 aromatic clamp remained stationary, holding the Glc moiety in the +1 subsite.

GPathFinder simulations with W434H (Glc at the −1 subsite and laminaripentaose in the +1 and +2 subsites and bulk solvent), revealed that the Glc egress path was identical to that of WT (path 1) (Figs. 7b, 8; Supplementary Movie 2). Here, a toll-like Arg158-Asp285-Glu491 barrier also re-oriented, Asp285 changed its rotameric state, and as Glc advanced through the lateral cavity, the side-chain of Tyr253 re-orientated, and the Arg291−Glu220 salt bridge was broken – these events enlarged the lateral cavity volume and facilitated Glc egress from the −1 subsite to bulk solvent.

For the W434A mutant (Glc at the −1 subsite and laminaripentaose in the +1 and +2 subsites, and bulk solvent), GPathFinder identified a different Glc exit path (path 2) (Figs. 7c, 8; Supplementary Movie 3). In this mutant, Glc exited through the preformed active site opening after the non-conservative mutation of W434 into Ala, while the Gly210-Met220 loop adopted a position that was seen in WT. This was supported by the higher flexibility of the region containing W434A, which adopted an altered conformation resulting from the displacement of Trp430 that could otherwise block path 2. In this Glc egress path, the toll-like Arg158-Asp285-Glu491 barrier, Tyr253, and the Arg291-Glu220 salt bridge were not involved.

When laminaripentaose was removed from the +1 and +2 subsites (and bulk solvent) in WT, W434H, and W434A, we identified path 3 (we termed it vertical) for Glc displacement (Fig. 8), where Glc utilised the space vacated by laminaripentaose and exited via the Trp286/Trp434 aromatic clamp, and through the +2 subsite into bulk solvent. This path could only be available when no products or substrates are bound. This scenario for Glc egress is, however, only a hypothetical option, also challenging our previous study[22], where in native HvExoI, two Glc molecules were bound at the −1 and +1 subsites, effectively blocking path 3.

In summary, GPathFinder simulations revealed that path 1 is the preferred route in WT and W343H with bound (1,3)-linked hydrolytic products, and is independent of a substrate length, contrary to W434A, which utilises path 2, where the essential structural attributes of substrate-product assisted processivity (toll-like barrier, salt bridge, loop motions, lateral cavity) do not participate.

## Discussion

In nature, processive (non-dissociative) catalysis is omnipresent in enzymes that synthesise, modify, and degrade biopolymers such as nucleic acids, proteins, and polysaccharides, and has been described in nucleases, peptidases, proteases, proteasomes, and endo- and exo-acting polysaccharide hydrolases[22–25,28–31,38]. Processive polysaccharide hydrolases have certain common characteristics that stem from the architectures of their catalytic sites that dictate spatial arrangements governing substrate specificity and binding (including directionality) and the dispositions of catalytic residues to ensure a high efficacy of catalysis. Processive endo- and exo-acting cellulases[23,28,39,40] and exo-acting chitinases[24,25,29,31] are known to be driven by processive hydrolysis or substrate-assisted catalysis, and the GH3 exo-acting β-D-glucan glycohydrolases with a pocket-shaped active site (such as HvExoI) by substrate-product assisted processive catalysis[22]. The difference between these two types is in the role of a hydrolytic product, that in the latter group evokes the formation of a lateral cavity, which serves as a conduit for Glc egress to allow for the next catalytic cycle using the retaining catalytic mechanism (Fig. 9). This type of catalysis enables multiple events that proceed without an enzyme losing contact with oligo- or polymeric substrates, contrary to distributive catalysis, where the catalyst follows a conventional association-conversion-dissociation pattern.

The crucial role of substrate binding in HvExoI implies a pair of the aromatic Trp286 and Trp434 residues that form a clamp at the +1 subsite, which is positioned next to a pocket-shaped site at the −1 subsite, housing the Asp285 and Glu491 catalysts[1–3]. These residues, typically around 5.5-6.5 Å apart locked in a rigid configuration via a dense H-bond network, are responsible for the hydrolysis of oligo- and polymeric β-D-glucosides to release Glc from non-reducing ends of substrates (Figs. 1a, b, and 9). To understand the mechanism that drives the unidirectional motion of the Glc product in HvExoI during substrate-product assisted processive catalysis, and specifically the roles of Trp286 and Trp434, we analysed substrate specificity in the context of elementary steps of Glc movement. We created HvExoI mutants replacing Trp286 and Trp434, with Ala, Phe, Tyr, and His residues, and compared their substrate specificity, steady-state

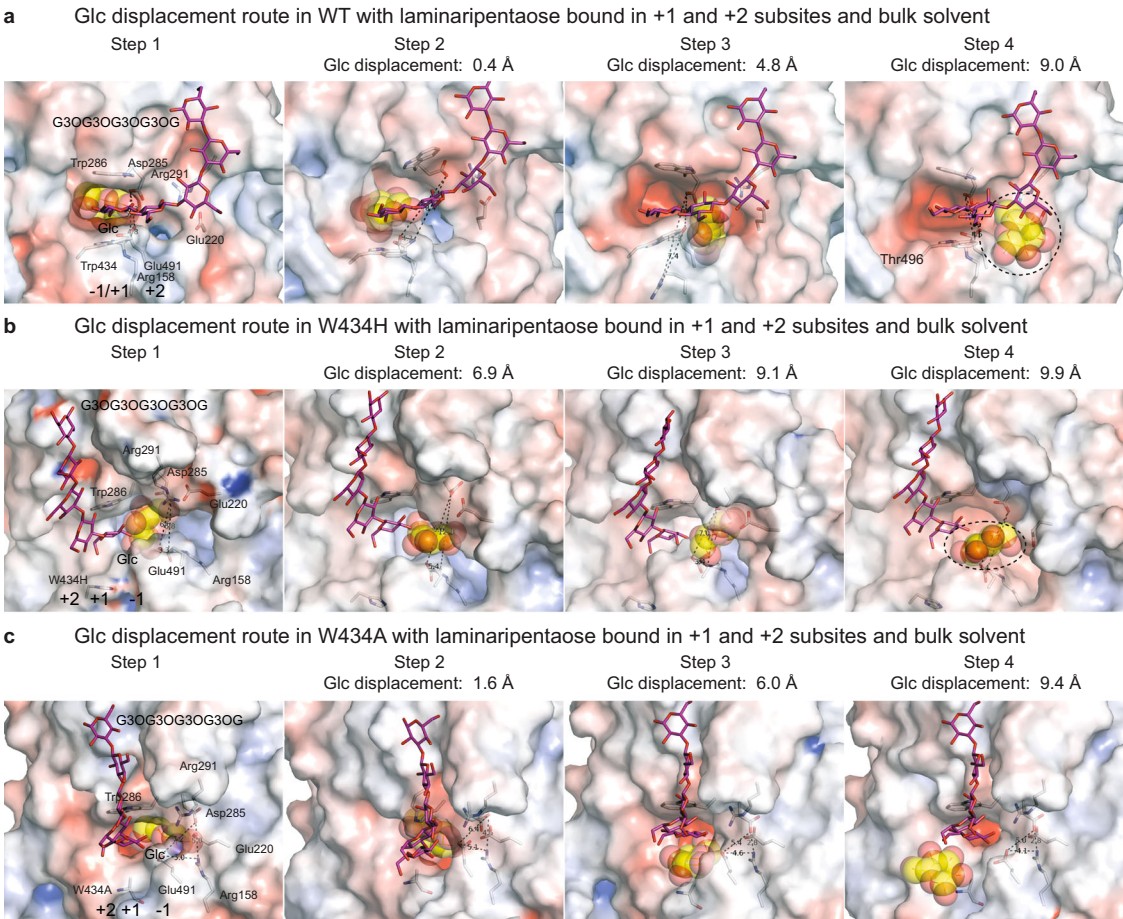

**Fig. 7 | Glc displacement routes in WT, and the W434H, and W434A mutants of HvExoI with Glc and laminaripentaose, calculated through cMD and GPath-Finder. a** WT; **b** W434H; **c** W434A – Four steps along Glc displacement routes based on converged structures in complex with Glc (−1 subsite; carbons in cpk yellow lines, and spheres) and laminaripentaose (+1 and +2 subsites and bulk solvent; carbons: magenta sticks) obtained by docking. Selected residues (carbons: atomic sticks), and positions of Glc, and laminaripentaose are indicated. Glc in the −1 subsite is separated from the lateral cavity that is evolving during Glc egress by the participation of the Arg158-Asp285-Glu491 toll-like barrier (triangles in dashed lines; distances indicated in Å). Surface morphologies of structures are coloured by electrostatic potentials: white, neutral; blue, +5 kT·e⁻¹; red, −5 kT·e⁻¹. Separations in Å between the positions of C1 carbons of Glc molecules, as they move from the −1 subsites in initial and final structures are indicated. The lateral cavity (**a**, **b**; indicated in black dotted ellipsoids) forms transiently, is partly exposed to bulk solvent, and facilitates Glc displacement. In panel **c**, Glc is displaced to bulk solvent through the opening formed by the W434A mutation.

kinetics, and inhibition parameters relative to WT. We conducted atomic resolution (1.5 Å – 2.5 Å) X-ray crystallographic analyses of 25 structures of WT and mutant forms in complex with positional isomers of thio-analogues, revealing the details of β-ᴅ-glucoside binding. We examined the conformational behaviour of thio-analogues and O-linked β-ᴅ-glucosides, and explored the binding and hydrolysis of the G4OG3OG natural substrate via cMD and QM/MM MD metadynamics simulations. To define the atomistic details and dynamics of reactant movements, we examined the unidirectional motion of Glc and its egress, and substrate sliding through the catalytic site or binding and displacements of laminaritriose and laminarihexaose hydrolytic products in the W434H and W434A mutants through cMD, GaMD, and GPathFinder simulations. These data were corroborated with phylogenomic analyses of GH3 family members, to trace the evolutionary origin of substrate-product assisted processivity, where β-ᴅ-glucan glucohydrolases are expected to retain an aromatic Trp286/Trp434 clamp. In the unrooted phylogenetic tree (Fig. 1c; Supplementary Table 1), we observed the clustering of plant β-ᴅ-glucan glucohydrolases to subclades 1-3 with both Trp residues conserved or diversified. We detected that through the evolutionary history of plants, β-ᴅ-glucan glucohydrolases with the aromatic clamp are absent in red alga *Porphyra* and unicellular green algae *Cyanobacterium* and *Chlamydomonas*, and amoeba

*Dictyostelium*, and have emerged with the evolution of land plants – these enzymes are present in liverworts and mosses (Embryophyte), ferns and conifers (Tracheophyte), a basal Angiosperm *Amborella trichopoda*, and dicotyledons (Eudicots) and monocotyledons (including grasses) (Fig. 1c). The β-ᴅ-glucan glucohydrolases with the Trp286/Trp434 signature are also richly represented in bacteria, e.g. the sludge *Chloroflexi* bacterium fermenting carbohydrates[41], and a plant-pathogenic *Xanthomonas* (Fig. 1c), similarly to other bacteria of the GH3 family[4] (*Bacteroides ovatus, Niveispirillum irakense, Pseudoalteromonas sp.* BB1), although β-ᴅ-glucan glucohydrolases from *Rhodothermus marinus* and *Saccharopolyspora erythraea* have the Tyr286-Trp434 signature[34].

During the investigations of physicochemical properties and substrate specificity in HvExoI and its mutants, we detected that the impact of Trp286 and Trp434 mutations on the hydrolysis of 4NP-Glc under a variety of pH and temperature conditions was relatively small compared to WT, except for W286H with its pH optimum shifted to the alkaline region (from 5.25 for WT to 6.7 for W286H) (Supplementary Fig. 1) due to a proximity effect to Asp285. However, the relative hydrolytic rates and specificities of mutants with (1,2)-, (1,3)-, (1,4)- and (1,6)-linked poly-, and oligosaccharides, and aryl-glycosides, were substantially altered. The W434A mutant lost its hydrolytic activities with polymers and oligomers, except (1,3)-linked oligosaccharides,

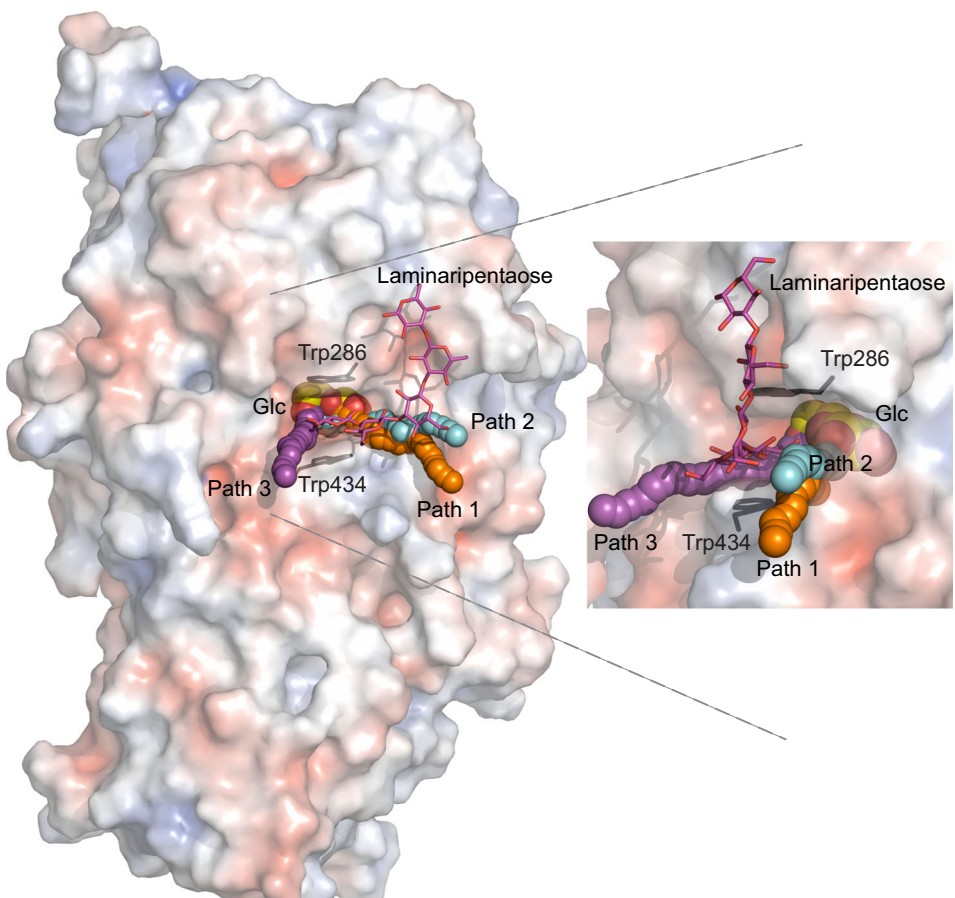

**Fig. 8 | Glc egress paths 1-3, illustrated on the top of WT HvExoI with bound Glc (carbons in cpk yellow lines, and spheres) and laminaripentaose (cpk magenta sticks).** Path 1 (WT and W434H), path 2 (W434A), and path 3 (vertical) are depicted in orange, cyan, and magenta spheres, respectively, and aligned next to each other. Path 3 is found when no ligand is bound at the +1 and +2 subsites. Illustrated trajectories represent frames that correspond to consecutive stages of Glc egress with conforming coordinates for Glc in each frame. Glc egress paths were identified by cMD simulations and GPathFinder. Surface morphology is coloured by electrostatic potentials: white, neutral; blue, +5 kT·e$^{-1}$; red, −5 kT·e$^{-1}$. Positions of Trp286 and Trp434 delineating the +1 subsite, are indicated in grey sticks. The right panel depicts the detailed view of paths 1-3, with HvExoI being rotated by around −80° via the y-axis relative to the left panel.

and acted as a strict (1,3)-β-D-specific β-D-glucosidase (Supplementary Table 2). In this mutant, no activity was observed with laminarin [(1,3)-β-linked polymer with (1,6)-β-branching at the 3:2 ratio[42]] suggesting that the binding of branched substrates may be influenced by steric effects. These modified hydrolytic rates in the Trp286 and Trp434 mutants were followed by apparent first- and second-order rate kinetic (Supplementary Table 3) and apparent inhibition constants (Supplementary Table 4). Changes in these parameters suggested that Trp286 and Trp434 play key roles in orienting positionally isomeric β-D-glucosides towards the formation of productive HvExoI:substrate complexes.

To address the atomic details underlying substrate binding and hydrolysis, the crystalline WT and mutant HvExoI forms were perfused with (1,2)-, (1,3)-, (1,4)- and (1,6)-linked thio-β-D-glucosides followed by X-ray crystallography analyses (Figs. 2–4; Supplementary Figs. 2–4; Supplementary Tables 5 and 6) to assess the contributions of residues that interacted with glycone/aglycone moieties and thus their involvements in catalysis. Our interpretations of the electron density maps of these complexes (25 in total) led us to postulate that it was primarily the aromatic clamp, that binds variously linked thio-glucosides at the +1 subsite, and mediates broad substrate specificity in HvExoI. In the bound (1,2)- and (1,6)-linked thio-glucosides, we observed $^4C_1$ conformations for Glc moieties at the −1 subsite, or $^4H_3$ or $^4H_3/^4E$ respective conformations for the (1,3)- and (1,4)-linked thio-glucosides (Supplementary Table 7), suggesting that certain

conformational flexibility of glucopyranose rings was present. This was supported by our previous observations that D-glucono−1,5-lactone (transition state analogue) with inherently a half-chair conformation was an effective inhibitor ($K_i = 2.1 \times 10^{-6}$ M)[21]. This is acceptable because the active site must be able not only to recognise the ground state of saccharides in $^4C_1$ conformation but also distorted conformations[43,44]. We concluded that the electron density maps observed in the complexes could represent snapshots of time-space averaged conformations, that are formed at different time intervals on the substrate-binding trajectory from the ground to transition states.

One significant observation stemmed from the comparative analyses in terms of binding of (1,2)-, (1,3)-, (1,4)- and (1,6)-linked thioglucosides by HvExoI (Figs. 2–4; Supplementary Figs. 2–4). In these WT complexes, the Glc moiety at the −1 subsite was oriented almost identically, contrary to the Glc moiety at the +1 subsite that adopted various orientations. In G3SG the apolar β face of Glc moiety was exposed to Trp286, while in the HvExoI:G2SG,:G4SG, and:G6SG complexes, the apolar thio-analogue sides mostly faced Trp434. It is known that in these stacking interactions, the saccharide surface encounters a side-chain aromatic moiety when an apolar face of a saccharide exposes three or more C–H groups close in space[45]. Concerning the substrate specificity from another angle, it is important to compare the G6SG position in enzyme forms. While the Glc moieties of G6SG at the −1 subsite were oriented nearly identically in WT and mutant

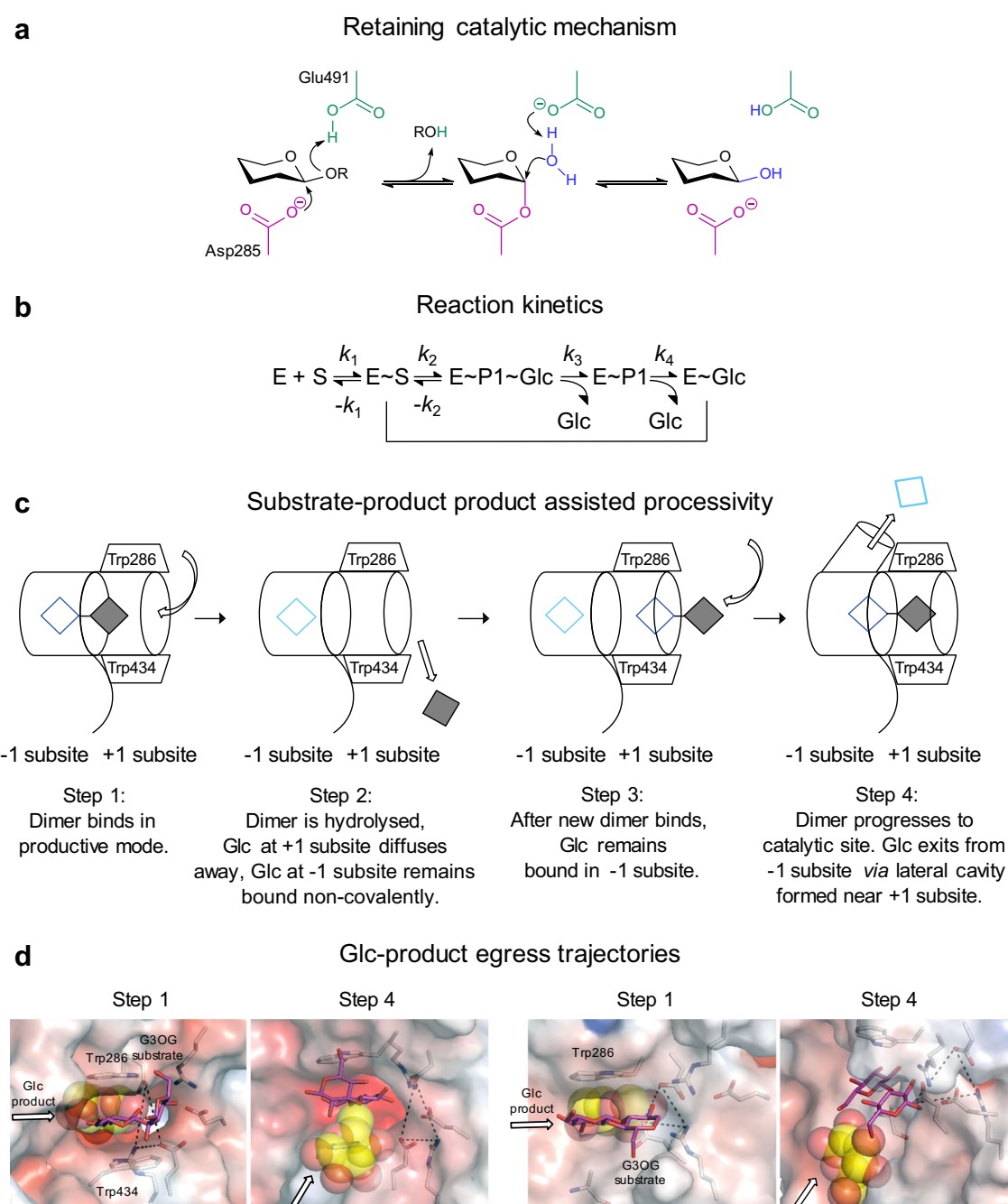

**Fig. 9 | Mechanistic view of the catalytic function of HvExoI, a family GH3 enzyme. a** HvExoI uses the retaining catalytic mechanism, where nucleophile Asp285 (magenta) and acid/base catalyst Glu491 (green) play key roles in substrate hydrolysis[1–3]. **b** Reaction kinetics indicates the formation of Michaelis complexes (E~S and E~S), and enzyme-product complexes before (E~P1~Glc) and following the first Glc egress (E~P1), and after hydrolysis of P1 (E~Glc)[1,2]. S and P1 are trimeric and dimeric molecules, respectively. First-order rate constants are shown. Entrapped Glc product from E~Glc is displaced after a new substrate is attached. The hydrolytic cycle with a (1,3)-linked substrate is repeated until it is hydrolysed to Glc (indicated by vertical and horizontal lines). **c** Simplified mechanism of Glc displacement with a disaccharide[22]. After the disaccharide (empty blue and filled grey

squares) bound in the −1 and +1 subsites (step 1) is hydrolysed and the reducing end Glc (aglycon) diffuses away from +1 subsite, the non-reducing end Glc (cyan square) remains non-covalently trapped. After a new dimer binds (step 3) and advances to the catalytic site, Glc modifies its binding patterns and egresses (large arrow) via a transient lateral cavity (cylinder in dotted lines) formed near the Trp clamp (step 4)[22]. Curved arrows indicate dimer entries into the catalytic site. **d** Glc displacement paths in WT (left panels), and the W434A mutant (right panels) with Glc and the G3OG dimer, calculated through cMD and GPathFinder. Large arrows indicate Glc positions at the initial (step 1) and after the final (step 4 – following Glc departure from −1 subsites) stages of catalysis (this work).

structures, the position of the Glc moiety at the +1 subsite varied – it was identical in WT and W286A, while in W434A it was rotated by 120° through the x-axis. This comparison underscored the pivotal role of Trp434 in correctly orienting saccharide substrates engaged in stacking interactions, also known as dispersion-driven C-H-π forces

(consisting of London dispersion forces)[46]. The geometries of these interactions with Trp, Phe, Tyr, and His involved face-to-face geometries, although the edge-face and offset stacked geometries also occur in protein complexes[47]. These stacking interactions are mediated by Trp residues in 40%, trailed by Tyr, Phe, and His[48].

To address the structural features of β-D-glucosides binding (or Michaelis complexes) in HvExoI at the atomic level, we compared the conformational behaviour of thio-analogues bound in crystalline forms (Supplementary Figs. 5 and 6) and after cMD simulations (Supplementary Fig. 8). These analyses revealed $^4C_1$ conformations of the glucopyranose moieties in the +1 subsite (Fig. 5; Supplementary Fig. 8; Supplementary Table 7), and the pseudo-$^4C_1$ conformation for the sugar moiety bound at the −1 subsite, although previous cMD analyses of a fungal GH3 enzyme did not find evidence of sugar distortion[47]. These findings supported the analyses of puckering conformations of glucopyranose rings in Michaelis complexes of other exo-acting GH enzymes[2,35,36], and suggested that these exo-acting enzymes differ from endo-acting ones, where substrate distortions were systematically observed[49–53].

Additionally, via cMD simulations and QM/MM MD metadynamics, we investigated the trajectory of the G3OG product originating from the G4OG3OG substrate, where the Glc moieties in $^4C_1$ conformations and FEL maps shared properties (Fig. 6; Supplementary Figs. 10 and 11). The cluster analysis of the G3OG product in WT and W434A showed positions in markedly different proportions (Fig. 6b). This led us to conclude that the G3OG product could not overcome a high energy barrier to achieve the productive binding mode, due to an unfavourable orientation of the Glc moiety in the −1 subsite. Here, the CH₂OH substituent faced the acid/base Glu491 residue, while the anomeric carbon was positioned away from the Asp285 nucleophile. A change in the ring puckering of the Glc moiety could eventually cause the anomeric carbon to approach Asp285, but such a possibility was eliminated based on the calculated FEL map (Fig. 6a). In W434A the G3OG product disposition allowed more flexibility, but G3OG still faced a high energy barrier to achieving a productive binding mode.

These findings have profound implications for the hydrolysis of mixed-linkage β-D-gluco-oligosaccharides by HvExoI, where the G4OG3OG natural substrate cannot slide through the active site from the +2 and +1 to the +1 and −1 subsites, although it could utilise the lateral cavity exit mechanism for Glc egress. The binding and hydrolysis of the G4OG3OG natural substrate for HvExoI proceed in two steps: (i) after the 1st glycosidic linkage (G4OG) from the non-reducing end is cleaved off, and Glc egresses through the lateral cavity; (ii) the G3OG product diffuses out of the active site and re-rebinds in the correct orientation to form a productive enzyme:substrate complex. This distributive catalytic process can occur with trimeric or longer substrates that are (1,2)-, (1,4)-, (1,6)- or (1,4;1,3)-linked, but not with (1,3)-linked substrates, which utilise both assets of processivity, i.e. Glc egress and substrate single-chain sliding movement forward through the catalytic site without dissociation. Notably, in HvExoI, the processive character is paramount to the catalytic efficiency and is correlated with high $k_{cat}$ values of the (1,3)-linked substrates (Supplementary Table 3). Similar (i-ii) steps can be used by G4OG4OG3G, which with G4OG3OG are the key low-molecular hydrolytic products generated from mixed-linkage (1,4;1,3)-β-D-glucans – polysaccharide components in cereal cell walls[5,6,54].

We alluded to the pivotal role of Trp434 in HvExoI that through stacking interactions participated in the orientations of substrates to positions amenable for hydrolysis through substrate-product assisted processivity. To extend our knowledge of the role of Trp434 in these processes, and why the evolution converged to Trp434, we applied cMD, GaMD[55], and GPathFinder[37] simulations, where cMD simulations safeguard the stability of complexes in geometries close to those in crystal structures, GaMD provide an enhanced sampling of conformational spaces, and GPathFinder explores dynamic traits of reactants binding or unbinding, via backbone motions and side-chain optimisations of discrete rotameric states. For these studies, we selected two mutants, which retained broad substrate specificity

(W434H), or behaved as a strict (1,3)-β-D-glucosidase (W434A). Summarising these findings, we revealed that: (i) the stability and time occupancy of the laminaribiose or laminaripentaose products in the subsites +1 and +2 were affected by W434H and W434A mutations; (ii) in WT and W434H, the lateral Glc egress path was used during substrate-product assisted processivity (path 1; Fig. 8); (iii) in W434A, Glc exited through the W434A-Trp286 clamp and a gap formed by the non-conservative W434A mutation, and without engaging substrate-product assisted processivity (path 2; Fig. 8); (iv) in WT and both mutants, Glc egressed through the aromatic Trp286/Trp434 clamp, when no oligosaccharides are present at this site (path 3; Fig. 8), which was energetically most favoured; (v) substrate-product assisted processivity can only be utilised, when the unbinding of the product at the +1 subsite is slower than Glc egress through the lateral cavity (Figs. 7–9; Supplementary Fig. 14; Supplementary Movies 1–3). Finally, our unconstrained MD simulations of HvExoI complexed with laminaripentaose, and Glc removed from the −1 subsite revealed that the sliding of laminaripentaose proceeded spontaneously, even in W434H or W434A, meaning that all enzymes position Glc moieties in the active sites correctly due to the structural complementarity.

Importantly, the combined experimental and computational approaches addressed the key role of the evolutionary conserved Trp286/Trp434 clamp in the GH3 family, which secures Glc unbinding and through C-H-π interactions guarantees the optimal binding of hydrolytic products while preventing their attachments by reducing-terminals. These approaches also revealed another important function, which is to assure that when 1,3-β-D-linked polymeric substrates bind to the active site and are hydrolysed with Glc egressing through an ad-hoc formed lateral cavity, the hydrolytic products could slide through the catalytic site and are rectified via substrate-product assisted processivity to warrant a next hydrolytic event. This process is accompanied by a cork-like motion between domains 1 and 2[22], which associates with an active site opened and closed states (Supplementary Movies 1–3).

Our studies provide a blueprint for emerging studies of reactant movements to advance the knowledge of catalysis in hydrolases and demonstrate that specificity and efficiency could be engineered via residue alterations that dock substrates into active sites. Particularly, the work through our multiscale strategy contributes to the understanding of catalysis in GH3 exo-hydrolases by defining structural trajectories and mechanisms for the breakdown of oligo- and polysaccharides (Fig. 9). We suggest that our findings could be utilised in enzyme design, as they are correlated with the statistical energy inferred from a maximum entropy model of homologous sequences[56]. Further, this knowledge can be exploited in the biomolecular design of artificial molecular motors[57,58], and more broadly in biotechnology and drug design, where enzymes are sought as environmentally friendly and cost-effective alternatives in bioremediation, and applications in medicine and pharmaceutics.

## Methods

### Generation of HvExoI mutants, protein expression, and purification

Mutant HvExoI forms were prepared by site-directed mutagenesis using the pPICZαBNH₈ vector containing an optimised cDNA fragment encoding mature HvExoI as a template[22,59,60]. Trp286 and Trp434 were singly replaced with alanine (Ala), histidine (His), phenylalanine (Phe), and tyrosine (Tyr). Doubly mutated W286F/W434F and W286F/W434A forms were also prepared to yield a total of ten mutants. Site-directed mutagenesis was carried out as described in the QuikChange® Site-Directed Mutagenesis Kit (Agilent Technologies, Santa Clara, CA, USA) using primers specified in Supplementary Table 8. DNA constructs were transformed in Pichia pastoris strain SMD 1168H to express and purify proteins as described[22,59,60].

## Enzyme stability, substrate specificity, kinetic analyses, and inhibition

Dependency of activities of WT and mutant HvExoI on pH (McIlvaine buffers in the pH range of 3.5–8.5 at 30 °C) and temperature (100 mM sodium acetate buffer, pH 5.25, temperature range 0–80 °C, 15 min incubations) were determined using 0.2% (w/v) 4NP-Glc[21,59]. Relative rates of hydrolysis of β-linked poly-, oligosaccharides, and aryl-glucosides were determined using near-homogenous HvExoI at 50 pmoles. Apparent first-order rate parameters ($K_M$ and $k_{cat}$) were determined with (1,3;1,6)-β-D-glucan (laminarin), barley (1,4;1,3)-β-D-glucan, laminaribiose, cellobiose, and 4NP-Glc with substrate concentrations ranging between 0.3-3 times $K_M$ values in duplicate as described[59,61]. Apparent inhibition constants of enzyme:inhibitor complexes ($K_i$) using the 4NP-Glc substrate were determined at six concentrations with G2SG-OMe, 4NP-G3SG, G4SG-OMe, and G6SG-OMe at 0.4-3 times the $K_i$ values in duplicate as described[21,22]. Kinetic parameters were calculated from the analyses of Michaelis-Menten plots with Grafit 7.0.3[62] and processed by a proportional weighted fit, using nonlinear regression[63]. The relative rates of hydrolysis, kinetic, and inhibition data were rounded to one or two decimal points (Supplementary Tables 2–4).

## Protein crystallisation and ligand perfusion

Near-homogenous WT and mutated HvExoI were crystallised by macro- and cross-seeding with native HvExoI microcrystals[22,60,64]. Briefly, crystallisation was carried out at 4 °C in handing drops consisting of 4-6 µL of HvExoI (4.5–12 mg/mL) and 4 µL of precipitant solution A (100 mM HEPES-NaOH buffer, pH 7.0 containing 2.3% (w/v) polyethylene glycol 400 and 1.6 M ammonium sulfate). Drops were placed on 22 mm siliconised circular glass coverslips, into which three to four 5–10 µm crystals of native HvExoI[64] were transferred by a cat whisker. Coverslips were suspended over 1 mL of mother liquor solutions (1.7 M ammonium sulfate in 46.9 mM HEPES-NaOH buffer, pH 7.0). Crystals of mutants grew within 10-28 days at 4 °C to 100-400 µm sizes in the longest dimension. Thio-analogue and Glc perfusion at saturating concentrations (list and conditions summarised in Supplementary Table 5) was carried out as described[22]. Briefly, crystals were perfused with thio-analogues or Glc dissolved in precipitant solution A for 5-720 min. After various perfusion times at 4±2 °C or ambient temperature, crystals were cryo-protected with 15% (v/v) glycerol in solution A and mounted on goniometers.

## Crystal structure determination

The X-ray diffraction data, except the WT:4NP-G3SG complex (Supplementary Table 6), were collected at the MX1 beamline of the Australian Synchrotron at 100 K (Oxford Instruments, UK) with a collimating mirror, double-crystal Si(111) monochromator on 03BM1 dipole/bending magnet, and ADSC Quantum 210r Detector. X-ray data of the WT HvExoI-4NP-G3SG complex were collected at 100 K (Oxford Instruments, UK) on an in-house Rigaku MicroMax-007HF with a second-generation true microfocus rotating anode generator, operating at 40 kV and 50 mA, Osmic VariMax optics for Cu, $K_\alpha$, and the R-AXIS IV++ Imaging Plate detector. All data were collected at 0.5-1° oscillations throughout 180-720°. Data were processed with the DENZO/SCALEPACK HKL 2000[65]. Autoindexing determined that space groups of crystals that were consistent with a tetragonal space group P4₃2₁2 in all instances. The structures were refined using CCP4 REFMAC5[66] and Phenix[67]. The previously solved structures of native HvExoI: PDB 1EX1, 1IEQ, 1IEV, 1IEW, 1IEX, and 1J8V) without ligands, ions, glycerol, and water molecules were used as starting models[2,3,22]. The iterative model building using REFMAC5[66], MIFIT[68], and Coot[69] allowed tracing of all the residues in electron density maps. Following convergence in a standard model refinement, approximately 2% improvements in the $R_{work}/R_{free}$ factor ratios were achieved by refining

domains (domain 1, residues 1-357; domain 2, residues 374-559) as two independent anisotropic domains with translation-libration-screw (TLS) motion[70]. Electron density maps of ligands were defined in the active site regions at 3σ levels in the m|$F_o$|-|$F_c$| maps (m is the figure of merit and D is an estimated coordinate error) and the water molecules were located automatically with CCP4 ARP at levels higher than 2σ levels. Water molecules were retained if they satisfied H-bond criteria and if the 2 m|$F_o$|-D|$F_c$| electron density maps were confirmed. During model building and refinement, 5% of the data were flagged for cross-validation using $R_{free}$ statistics[71,72]. PROCHECK[73] was used to check the stereo-chemical quality of models. Ramachandran plots of structures showed that 99.8% of the residues were found in the most favourable, additionally allowed, and generously allowed regions of the plot, with the well-defined density map for Ile432, the only residue located in disallowed regions. Structural images were prepared in PyMOL Molecular Graphics System v2.3.3 (Schrödinger LLC, Portland, OR, USA). The summary of 25 structures with thio-analogues and Glc and PDB accession numbers, data collection, and refinement statistics are reported in Supplementary Table 6. The (2 m|$F_o$|-D|$F_c$|) difference electron density maps were calculated using Phenix[67] with the mtz and PDB files after CCP4 REFMAC5 refinements. The maps were contoured at the 1.0 σ level. Ligands bound in active sites (Figs. 2–4; Supplementary Figs. 2–4) were visualised in PyMol Molecular Graphics System v2.3.3.

## cMD simulations of Michaelis complexes derived from thio-ligand complexes

A total of eight models were generated for the following systems: W286F-G2(S or O)G in complex with G2SG-OMe; PDB 6JG7; WT-G3(S or O)- in complex with 4NP-G3SG; PDB 6JG2; WT-G4(S or O)G in complex with G4SG4OG4S; PDB 6JG1, and WT-G6(S or O)G in complex with G6SG-OMe; PDB 3WLP. The procedure consisted of (i) reverting the mutation in the W286F crystal structure; (ii) removing SO₄²⁻ anions, glycerol (GOL), polyethylene glycol (PEG), N-acetyl glucosamine (NAG), and aglycon moieties of thio-analogues; (iii) neutralising the systems with three Na⁺ cations; and (iv) adding a cubic water box around proteins. For O-linked saccharide systems, an additional step was taken by replacing S-atoms in thio-analogues with O-atoms. The tleap code, included in AMBER 11[74] was chosen to parametrise systems using the ff99SB force field[75] for the protein, Glycam06[76] for saccharides, and the TIP3P force field[77] for water molecules. The equilibration protocol consisted of a minimisation step (8 ×10³ steps), progressive heating to 300 K (700 ps), NPT equilibration under a pressure of 1 atm to ensure physiological density (100 ps), and a final NVT equilibration (with a time step value of 2 ps using the SHAKE[78] algorithm to constrain bonds involving hydrogens) until the RMSD values of protein backbones stabilised. Default parameters were used for non-bonding interactions and long-range electrostatics. Simulations at NVT were extended to 200 ns each.

## cMD simulations of the Michaelis complex with a natural substrate G4OG3OG bound to WT and W434A, based on the WT:G4SG4OG4S complex

The natural substrate G4OG3OG was modelled, based on the WT HvExoI crystal structure in complex with G4SG4OG4S (PDB 6JG1), where, S-atoms in G4SG4OG4S were replaced by O-atoms, the gly-cosidic linkage was re-arranged to be 3-O-linked between the 2nd and 3rd Glc moieties from the non-reducing end, and the atoms were carefully moved such that the system maintained β-anomeric conformation. Further, SO₄²⁻ anions, GOL, PEG, and NAG were removed, and the system was neutralised with three Na⁺ cations and solvated in a cubic water box. From this structure, two more systems representing the product of the hydrolytic reaction of G4OG3OG were modelled. The Glc product at the −1 subsite was removed in both

systems and for one system the WT structure was retained, while for the other, Trp434 was mutated to Ala. These three systems were parametrised using the tleap program of AmberTools21[79] using the ff19SB protein force field[80], Glycam06[76] for saccharides, and the TIP3P force field[77] for water molecules. Amber20[79] was used to minimise and equilibrate all systems using the protocol described above, and applying restraints when necessary until the RMSD values of protein backbones stabilised. Simulations were extended to 150 ns each.

## QM/MM MD metadynamics simulations of G3OG and G4OG3OG substrates in WT

For these studies or conformational Free Energy Landscape (FEL) simulations of Glc moieties of G3OG and G4OG3OG positioned in active sites of HvExoI, we used the CPMD code (parallelised plane wave/pseudopotential implementation of Density Functional Theory, Copyright IBM Corp 1990-2008)[81] patched with the open-source community-developed PLUMED2 library[82,83]. For the conformational FEL simulation of the G3OG disaccharide, the QM box (region) included 45 atoms corresponding to the disaccharide in the −1 and +1 subsites; the MM region included the protein and water molecules (85,478 MM atoms). For the conformational FEL simulation of the G4OG3OG trisaccharide, the whole sugar was not included in the QM box (region) but instead, the Glc molecule in the −1 subsite and the half-ring of Glc in the +1 subsite (with two monovalent pseudopotentials to saturate the QM region) were included in the QM box, corresponding to 29 QM atoms in total. The other half-ring of the Glc molecule in the +1 subsite and the third Glc moiety placed in the +2 subsite were described as MM atoms, which also included protein and water molecules (11,5415 MM atoms)[84] (Supplementary Fig. 15). The G3OG and G4OG3OG systems were enclosed in isolated supercells of 13.5 Å x 19.3 Å x 14.8 Å and 13.3 Å$^3$ sizes. Kohn-Sham orbitals were expanded on a plane wave basis set with a kinetic energy cut-off of 70 Ry. Troullier-Martins ab initio pseudopotentials[85] were used for all elements. The PBE functional[86] in the generalised gradient-corrected approximation of DFT with a time step of 0.12 fs and a fictitious electron mass of 600 a.u. for the Car-Parrinello Lagrangian were used in both systems. The Nosé-Hoover thermostat[87] was coupled to the systems to reach a constant temperature of 300 K. After the systems were equilibrated, metadynamics simulations were initiated[88]. To obtain conformational FEL of Glc at the −1 subsite in both systems, we used the polar Cremer and Pople puckering coordinates[89], as φ and θ collective variables. The well-tempered metadynamics algorithm[90] was used with a bias factor of 12, a temperature of 300 K, and an initial height of Gaussians of 0.75 kcal/mol. The Gaussian width was 0.1 rad with a deposition time of 400 molecular dynamics steps (corresponding to 48 fs). The simulations were stopped after a free energy error was below 0.6 kcal/mol (analysed using the block error analysis[91]), corresponding to 2822 Gaussians (140 ps) and 2980 Gaussians (150 ps) for G3OG and G4OG3OG, respectively (*cf*. Supplementary Fig. 11).

## Analyses of cMD and QM/MM MD metadynamics simulations

Data analyses were performed in VMD1.9.3[92] using home-made Python3 scripts, the matplotlib library[93], and the previously described script[94]. The open-source, community-developed PLUMED library[83] version 2.7.2[82] was used for block analyses. The cpptraj tool from Ambertools21 was used to obtain representative structures of the most populated clusters in simulations.

## cMD simulations of laminaritriose or laminarihexaose hydrolysis products by WT and the W434H and W434A mutants

For each enzyme form, a model was built with bound hydrolysed products from laminaritriose (*i.e.* Glc at the −1 subsite and G3OG at the +1 and +2 subsites) or laminarihexaose (Glc at the −1 subsite and

laminaripentaose at the +1 and +2 subsites and pointing to bulk solvent). For a comparison, models for the apo and laminaritriose substrate-bound forms of HvExoI were also generated. All models were built based on the crystal structures of WT (in complex with 4NP-G3SG; PDB 6JG2), W434A (in complex with 4NP-G3SG3OG; PDB 6L1J), and W434H (in complex with 4NP-G3SG; PDB 6JGN). The starting structures for cMD simulations were prepared using USFC Chimera[95]. Ion, water, GOL, and PEG molecules present in crystal structures were removed and proteins were protonated at pH 7, and with the catalytic acid Glu491 in a neutral form as it corresponds to this step of the catalytic cycle[1,2]. The (1,3)-linked β-D-glucosides (laminaribiose, laminaritriose, and laminaripentaose) used in simulations were generated in the carbohydrate builder GLYCAM, using the legacy version of the software (https://glycam.org/cb/legacy). To ensure that the starting structures had Glc bound in the −1 subsite and the substrates non-reducing ends occupied the +1 and +2 sites, oligosaccharides were spatially matched to substrates present in crystal structures of WT and mutant HvExoI using the match commands in USFC Chimera. System solvation, coordinate, and topology file acquisitions (prmtop and inpcrd) were performed in tleap, included in the AMBERTools21 suite[96]. Systems were solvated in TIP3P water molecules and neutralised with Na$^+$ ions, using a cubic box, with standard periodic boundary conditions set. The system was parameterised using the Amber ff99SB-ILDN force fields[97] for proteins, GLYCAM_06j[76] for the carbohydrates, and the TIP3P[77] for water molecules. Simulations were carried out in AMBER 18[98] with CUDA acceleration. The cMD simulation protocol consisted of: (i) initial minimisation (10$^5$ steps); (ii) heating (100 ps, with protein backbones restrained); (iii) NVT equilibration (100 ps, with restrained backbones); (iv) NPT equilibration (500 ps, unrestrained); and (v) NPT production at 300 K under a pressure of 1.0 atm. Every simulation was carried out with a time step value of 2 ps using the SHAKE[78] algorithm to constrain bonds involving hydrogens. A cutoff value of 14.0 Å was used for non-bonding interactions and the Particle Mesh Ewald method for long-range electrostatics[99]. Production simulations were run for 1000 ns in all cases except laminarihexaose, where we applied 600 ns simulation time. Due to the size of laminaripentaose, we also conducted GaMD simulation to enhance the sampling of the conformational space adopted by proteins and ligands.

## GaMD simulations of laminaritriose and laminarihexaose hydrolysis products by WT, and the W434H and W434A mutants

For GaMD simulations[55], AMBER 18[98] was used with the same forcefield described in the previous paragraph. The GaMD protocol consisted of an initial equilibration stage, where the potential boost was applied and boost parameters were updated, and production runs were updated with fixed boost parameters. A dual boost on both dihedral and total potential energy was applied (igamd = 3). As a starting point, the structures obtained after clustering in Chimera cMD trajectories were used. For W434H and W434A, clustering was performed on the first part of the trajectory since the oligosaccharides did not remain bound in the +1 and +2 subsites for extended time periods, and GaMDs production simulations were run for around 800 ns. For comparison, GaMD simulations of bound laminaritriose and the apo form were also examined using 400 ns production runs. GaMD inputs were generated following the recommendations from the developers (Y. Miao and R. Walker, Gaussian accelerated molecular dynamics tutorial; https://miaolab.ku.edu/GaMD/lib/GaMD_Amber-Tutorial.pdf). After the simulations were completed, mdcrd files were converted into a dcd format using the Amber Cpptraj tool and visualised in USCF Chimera. All systems were checked for convergence[100–102] by calculating RMSD, all-to-all RMSD, and principal component analyses values for each set of simulations with a Python script.

## GpathFinder simulations to explore Glc displacement routes in WT, and the W434H and W434A HvExoI mutants

This method uses the multi-objective genetic algorithm of the GaudiMM engine[103], wherein genes are structural operators (e.g. atomic positions and displacements), and objectives (fitness functions in evolutive algorithms) are structural and energetic descriptors (e.g. force field functions, distances, angles, etc.). The GaudiMM engine combines full-atom representations of a ligand and a protein and allows diverse levels of structural flexibility including internal rotation, global rotation and translation of ligands, and the exploration of protein motions that are in a short-range (e.g. internal rotation, global rotation, and translation of ligands) or a long-range (e.g. rotameric exploration and normal mode analysis of the protein). Typically, in GpathFinder runs, fast energetic evaluations are needed, which are provided by a docking scoring function, *i.e.* using Autodock Vina[104] and steric clashes. GpathFinder's philosophy relies on the conformational selection hypothesis, as collective motions (provided by the sampling of the lowest energetic modes obtained from Normal Mode Analysis) are responsible for conformational changes of proteins associated with their function, including ligand binding. Ligand diffusion of Glc from the −1 subsite was calculated with GPathfinder for six systems: WT, W434H, and W434A with or without laminaribiose (i-iii) or laminaripentaose (iv-vi) hydrolysis products bound in the +1 and +2 subsites and pointing to bulk solvent for laminaripentaose, since the Glc molecule occupied the −1 subsite. In each case, the starting structure calculations were chosen from the most representative clusters of cMD simulations described above, and GpathFinder calculations were performed using standard genetic algorithm parameters[37]. Clashes and the scoring Autodock Vina function were used as objectives, together with genes representing rotamers and Normal Modes Parameters calculated with ProDy[105]. The pool of individuals consisted of a population of sixteen candidates that satisfied the genetic algorithm for 800 generations. Calculations were analysed using GaudiView for visual inspection, together with the movie plug-in in UCSF Chimera to inspect trajectories. Other analyses were performed through in-house UCSF Chimera built-in plug-ins, such as finding clashes/contacts and distances, to locally explore interactions of Glc with amino acid residues during egress pathways.

## Unconstrained MD simulations to explore sliding of substrates in WT, and the W434H and W434A HvExoI mutants

These simulations proceeded with laminaripentaose in the +1 and +2 subsites and pointing to bulk solvent, and Glc removed from the −1 subsite. The systems were generated from the structures obtained from GaMD simulations and were run as described above for ternary complexes, with production runs of 200 ns each.

Additional methods are available in Supplementary Methods (Materials, Phylogenetic analysis, Organic Syntheses of 4NP-G3SG and G4SG-OMe).

### Reporting summary

Further information on research design is available in the Nature Research Reporting Summary linked to this article.

## Data availability

The data that support this study are available from the corresponding author upon request. The atomic coordinates and structure factors have been deposited in the Protein Data Bank (www.pdb.org) with the following PDB accessions: WT HvExoI, and in complex with 4NP-G3SG, and (G4SG4O)₃ are 6JG2, and 6JG1, respectively. The PDB accession of W286A in complex with G6SG-OMe is 6JG6, and W286F in complex with G2OG-OMe, 4NP-G3SG, and G6SG-OMe are 6JG7, 6JGA, and 6JGB. The PDB accession of W286Y in complex with Glc, and G6SG-OMe are 6JGC and 6JGD. The PDB accession of W434A in complex with Glc, G2SG-OMe, 4NP-G3SG3OG, (G4SG4O)₃, and G6SG-OMe are 6KUF,

6JGE, 6LIJ, 6LBB, and 6K6V. The PDB accession of W434F in complex with G2SG-OMe, 4NP-G3SG, (G4SG4O)₃, and G6SG-OMe are 6JGG, 6LC5, 6JGK, and 6LBV. The PDB accession of W434H in complex with G2SG-OMe, 4NP-G3SG, G4SG-OMe, and G6SG-OMe are 6JGL, 6JGN, 6JGO, and 6JGP. The PDB accession of W434Y in complex with G2sG-OMe, 4NP-G3SG, G4SG-OMe, and G6SG-OMe are 6JGQ, 6JGR, 6JGS, and 6JGT. Source data are provided with this paper.

## Code availability

Software applications described in Methods with associated references were used without code modifications. For analyses of geometrical parameters and MD simulations, we used in-house scripts, which are available on GitHub (https://github.com/insilichem/utils/; https://github.com/roviralab/utils) and ZENODO repositories[106,107].

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

## Acknowledgements

Funding was supported by the Australian Research Council (DP120100900) to M.H., and the Australian Synchrotron to M.H. and V.A.S. J.G.S. and M.H. thank the University of Adelaide and M.H. acknowledges the Huaiyin Normal University for support. J.R.K.C. and S.L. acknowledge funding from the Suranaree University of Technology, the Synchrotron Light Research Institute, and the Thailand Research Fund (RSA6280073). A.N.H. and C.R. acknowledge funding from the Spanish Ministry of Science, Innovation, through MICINN/AEI/FEDER, UE, PID2020-118893GB-100, the Spanish Structures of Excellence María de Maeztu (MDM-2017-0767), and the CARBOCENTRE project, which is funded by the European Research Council under the European Union's Horizon 2020 research and innovation program ERC-2020-SyG-95123. C.R. and A.N.H. also thank the Barcelona Supercomputing Center and Red Nacional de Supercomputación RES for computational resources at Marenostrum IV and CTE-Power for technical support. S.A.-G thanks the Spanish Ministry of Economy and Competitiveness for a Predoctoral Scholarship (BES-2012-051782). S.P. and S.F. recognise Glyco@Alps via ANR-15-IDEX-02, Carnot Institut PolyNat, Labex ARCANE, and CBH-EUR-GS through NR-17-EURE-0003, and thank NanoBio ICMG (UAR 2607) for providing expertise for mass spectrometry and NMR analyses. J.-D.M., X.F.L., and L.M. acknowledge the Spanish Ministerio de Ciencia, Innovación y Universidades via the PRE2019-088412 fellowship, and the PID-2020-116861GB-I00 and PGC2018-098592-B-100 grants, and L.M. acknowledges the Universitat Autònoma de Barcelona Talent Program. The use of the Australian synchrotron was supported by the Australian Synchrotron Research Program, which is funded by the Commonwealth of Australia under the Major National Research Facilities Program.

## Author contributions

S.P. and S.F. synthesised thio-ligands; S.L. and J.R.K.C. implemented mutagenesis; S.L. completed enzyme kinetics; S.L. and M.H. purified and crystallised proteins; V.A.S. and M.H. collected X-ray data; S.L., V.A.S., and M.H. refined crystal structures; M.H. and J.G.S. performed phylogeny analyses; A.N.H., S.A.G., and C.R. performed cMD, and QM/MM MD metadynamics calculations; X.F.L., J.-D.M., and L.M. implemented cMD, GaMD, and GPathFinder simulations; X.F.L., A.N.H., and M.H. generated animations and structural images. M.H. wrote the paper with contributions from S.L., S.F., X.F.L., A.N.H., J.-D. M., L.M., and C.R. M.H. oversaw the project.

## Competing interests

The authors declare no competing interests.

## Additional information

¹School of Agriculture, Food and Wine, and Waite Research Institute, University of Adelaide, Waite Research Precinct, Glen Osmond, SA, Australia. ²Department de Química, Universitat Autònoma de Barcelona, Bellaterra, Spain. ³Department of Química Inorgànica i Orgànica and Institut de Química Teòrica i Computacional, Universitat de Barcelona, Barcelona, Spain. ⁴The Florey Institute, University of Melbourne, Victoria, Australia. ⁵School of Chemistry, Suranaree University of Technology, Nakhon Ratchasima, Thailand. ⁶Université Grenoble Alpes, Centre de Recherches sur les Macromolécules Végétales, Grenoble, France. ⁷Institut de Biotecnologia i de Biomedicina, Universitat Autònoma de Barcelona, Bellaterra, Spain. ⁸Institució Catalana de Recerca i Estudis Avançats, Barcelona, Spain. ⁹Jiangsu Collaborative Innovation Centre for Regional Modern Agriculture and Environmental Protection, School of Life Science, Huaiyin Normal University, Huai'an, China. ¹⁰These authors contributed equally: Sukanya Luang, Xavier Fernández-Luengo, Alba Nin-Hill. ✉e-mail: maria.hrmova@adelaide.edu.au

