## [Peer Review File · Nature Communications]

The evolutionary advantage of an aromatic clamp in plant family 3 glycoside exo-hydrolasesREVIEWER COMMENTS

Reviewer #1 (Remarks to the Author):

The manuscript by Sukanya Luang et al. entitled “The evolutionary advantage of an aromatic clamp in plant family 3 glycoside exo-hydrolases” elegantly describes an in depth biochemical, structural and mutational study combined with molecular modelling to decipher the mechanistic importance of structural features in the active site of plant GH3 enzymes for processivity, highlighting that of a conserved Trp-clamp. The study also includes phylogenomic analyses to question the evolutionary advantage of the presence of this Trp-clamp. These studies are crucial for understanding the finetuned structure-function relationships leading to processivity of these enzymes, a mode of action which remains difficult to assess. With a tremendous amount of work the authors show, as a major result, that processivity was sensitive to mutation-specific alterations of the Trp clamp, and that a lateral cavity for glucose displacement and sliding of (1,3)-linked hydrolytic products through the catalytic site was important for efficiency. The authors conclude on the key role of the evolutionary conserved Trp286/Trp434 clamp in the GH3 family, and the importance to understand reactant trajectories and movements that are tailored by the enzymes structure, prior to designing and engineering new or adapted ones.

Globally, the manuscript is well written and comprehensive and the hypotheses drawn from the data are well argued and supported by the results. The study is thoroughly conducted and perfectly meets with expected standards. The study is within the scope of the journal, and contains original and significant results worth of publication and that will be of interest to a large community. I have listed below only a view very minor comments and corrections that might need to be considered prior to acceptance.

Comments :

1. Introduction, page 2 line 64 “Several three-domain architectures consisting of an (α/β)₈ barrel, and (α/β)₆ sandwich folds,...” Only two of the three domains are named here;

I believe the authors meant “Several three-domain architectures consisting of an (α/β)₈ barrel, a (α/β)₆ sandwich fold and one β -sandwich domain,... ”

2. Figure 1a. The side view of the Glc molecule in the -1 binding site does not allow to see the precise orientation; it would be helpful if the Glc-OH groups (at least O1 and O4 or O6) were labeled in the figure. Also, it would help the reader if the Figure legend would include the indication by which angle the structure is rotated between 1a and 1b.

3. Discussion, page 12 lines 475-477; The sentence (or sentences) are difficult to read due to a missing bracket and full stop.

Reviewer #2 (Remarks to the Author):

In this manuscript the authors present a hypothesis on how an evolutionary conserved Trp286/Trp434 clamp in the GH3 family facilitates processive hydrolysis of substrates. This hypothesis is backed up by data from biochemical assays, X-ray crystallography studies and computational chemistry. In the crystallographic approach a multitude of enzyme variants and substrate analogs have been thoroughly analyzed. The computational approach is broad, and the authors have selected suitable methods to investigate interactions between substrates, products, and enzyme variants. In this manuscript the authors have combined the powerful methods of structural biology and computational chemistry and have thus enabled analysis of molecular mechanisms at both at the macromolecular nano-scale to the dynamics of the electronic structures in the active site. This is an original and noteworthy contribution to the field.

Comments to the authors:

The amount of data presented in this manuscript is huge with several multipanel figures. I would suggest adding an overview figure where binding, hydrolysis and product release processes are illustrated in simpler terms to help the reader catch on.

For the classical MD simulations there is no mention of the thermostat employed, this should be included. While the setup/equilibration for the cMD simulations described in lines 671-700 is accurately described, this is not the case for the cMD simulations described in lines 607-635.

For the three different sections describing cMD simulations (starting at line 607, 621 and 671) different force fields have been applied (ff99SB, ff19SB with TIP3P instead of OPC water model, and ff99SB-ILDN). I would like the authors to comment on why different force field were applied in these simulations and if this has any consequences for the analysis of the results. For the GaMD simulations is not explicitly mentioned what force field that was applied.

In all figures containing electron density maps (both in the main manuscript and supplementary information) there are no difference maps included. This should at least be explained or the figures should be updated so that the reader can get an visual impression of the quality of the modeling. Also, in all the accompanying legends the term “ δ -level” is used while “ σ -level” is correct (change from delta-level to sigma-level).

In supplementary Table 6 R-merge should be included for the highest/outer shell and the CC1/2 parameter should be added for each structure.

Considering the text below on data availability, I would prefer that the scripts were made publicly available at e.g. github and that a DOI was included. As a recent paper demonstrated – the reasonable request do rarely work out in practice.

« For analyses of geometrical parameters and MD simulations, we used in-house scripts, which are available upon reasonable request from the corresponding author.»

Reviewer #3 (Remarks to the Author):

Reviewer report on " The evolutionary advantage of an aromatic clamp in plant family 3 glycoside exo-hydrolases"

Luang et al have undertaken a very thorough study on the mechanism of processivity on a GH3 exo-hydrolase. The manuscript is well written and major conclusions are supported by well-designed experiments. Still, there is a serious lack in the presentation of the history in GH processive mechanism. Especially on the role of aromatic amino acids. I feel it is urgent that readers of this manuscript are informed that some of the major thoughts and findings presented in this work has already been shown by other research groups previously. This said, Luang et al provides structural and computational evidence for the role of aromatic amino residues in the processive mechanism that are new, and hence, deserves to have their work in Nature Communications. If the authors are able to include the findings from previous studies as support for their work and as a basis for what is already known, I will recommend that the work is published.

Major points

- Line 78, Page 2. " Recently, we envisaged that β -D-glucan glucohydrolases, such as HvExoI, adopt so-called substrate-product assisted processive catalysis, where after an incoming substrate binds in a productive mode, the Glc product evokes the formation of a transient lateral cavity, which serves as a conduit for its egress to allow for the next catalytic round²¹." I have to say it is greatly disturbing to address this topic without citing papers such as Igarashi et al. Science 2011, Horn et al. PNAS 2006, and Zakariassen et al. J. Biol. Chem. 2009, which demonstrates that when the product dissociates, it is possible for a processive movement of the polysaccharide/oligosaccharide to occur.

- Line 84, Page 3. Again, it is well documented that aromatic amino acids are vital for the processive mechanism in GHs (Horn et al. PNAS 2006 and Jana et al. J. Phys. Chem. B. 2016).

- Line 116, Page 3. "Trp286/Tyr286 and Gln434/Tyr434/Trp434 in subclade 2, and Gly434/His434/Lys434/Trp434 in subclade 3 (Fig. 1c), which may impact substrate binding and processivity in these enzymes." The role of aromatic amino acids in the processive mechanism of GHs have clearly been demonstrated in Horn et al. PNAS 2006 and Jana et al. J. Phys. Chem. B. 2016. Moreover, the role of polar amino acids (non-aromatic amino acids) has previously been described by Hamre et al. Biochemistry 2015. The authors are not the first to propose this.

Minor points

- Line 50, Page 2. Would it be beneficial to introduce the Figure of HvExol already here as it is detailed important information of the structure and function?

- Line 56, page 2. "while the reducing Glc moiety at the +1 subsite is constrained weakly,". I understand that this is citations of previous works, but stacking of sugar moieties with aromatic amino acids are not weakly. These can be in the range of 2 to 5 kcal/mol (Zolotnisky et al. PNAS 2004, Jana et al. J. Phys. Chem. B. 2016).

- Line 153, Page 4. "These data signal that although the mutations of Trp434 exerted a stronger inhibitory trend compared to Trp286, both Trp residues are required for binding isomeric β -d-glucosides into the productive binding modes required for high catalytic efficiency." Is the word "inhibitory" appropriate as this is normally associated with enzyme inhibition?

- Line 158, Page 4. "to quantify dissociation constants (K_i)" K_d is the dissociation constant, K_i is the inhibition constant.

- Line 166, Page 4 and 5. "while the most effective thio-inhibitor was G6SG-OMe, except that it did not bind to W434A and showed around 6- and 10-fold lower K_i constant values with W434H and W434F, respectively (Supplementary Table 4)." Please address with a short line in the text why this is so.

- Line 185-186, page 5. "4NP-G3SG bound tightly to WT HvExol with the respective K_i constants of 382×10^{-6} M and 377×10^{-6} M for recombinant and native enzymes." Do you have three significant number in your experiments? This is not likely. Unless you report standard deviations, it should be 0.4×10^{-9} M. This goes as well to (line 176) K_i of $3,270 \times 10^{-6}$ M and many others. Check throughout the manuscript.

REVIEWER COMMENTS

Reviewer #1 (Remarks to the Author):

The manuscript by Sukanya Luang et al. entitled “The evolutionary advantage of an aromatic clamp in plant family 3 glycoside exo-hydrolases” elegantly describes an in depth biochemical, structural and mutational study combined with molecular modelling to decipher the mechanistic importance of structural features in the active site of plant GH3 enzymes for processivity, highlighting that of a conserved Trp-clamp. The study also includes phylogenomic analyses to question the evolutionary advantage of the presence of this Trp-clamp. These studies are crucial for understanding the finetuned structure-function relationships leading to processivity of these enzymes, a mode of action which remains difficult to assess. With a tremendous amount of work the authors show, as a major result, that processivity was sensitive to mutation-specific alterations of the Trp clamp, and that a lateral cavity for glucose displacement and sliding of (1,3)-linked hydrolytic products through the catalytic site was important for efficiency. The authors conclude on the key role of the evolutionary conserved Trp286/Trp434 clamp in the GH3 family, and the importance to understand reactant trajectories and movements that are tailored by the enzymes structure, prior to designing and engineering new or adapted ones.

Globally, the manuscript is well written and comprehensive and the hypotheses drawn from the data are well argued and supported by the results. The study is thoroughly conducted and perfectly meets with expected standards. The study is within the scope of the journal, and contains original and significant results worth of publication and that will be of interest to a large community. I have listed below only a view very minor comments and corrections that might need to be considered prior to acceptance.

Comments to the authors:

1. Introduction, page 2 line 64 “Several three-domain architectures consisting of an (α/β)₈ barrel, and (α/β)₆ sandwich folds,....” Only two of the three domains are named here; I believe the authors meant “Several three-domain architectures consisting of an (α/β)₈ barrel, a (α/β)₆ sandwich fold and one β -sandwich domain,.... ”

Thank you for pointing to the missing term. The missing “ β -sandwich fold” term (page 2, lines 67-68) has been added to the sentence.

2. Figure 1a. The side view of the Glc molecule in the -1 binding site does not allow to see the precise orientation; it would be helpful if the Glc-OH groups (at least O1 and O4 or O6) were labeled in the figure. Also, it would help the reader if the Figure legend would include the indication by which angle the structure is rotated between 1a and 1b.

Thank you for the suggestions. Panels in Fig. 1a and Fig. 1b have been re-drawn, and the C1 and C4 carbons of the glucose molecules located in the -1 subsite have been labelled in both images. Glucose molecules in both panels are now oriented identically. Further, the interacting residues with the glucose molecule in the subsite -1 have been unified in both panels, so the information is directly comparable. The legend in Fig. 1 was adjusted (page 27, lines 1103-1103).

3. Discussion, page 12 lines 475-477; The sentence (or sentences) are difficult to read due to a missing bracket and full stop.

We apologise. The missing full-stop has been re-introduced to the sentence, now line 494 (page 12, line 497).

Reviewer #2 (Remarks to the Author):

In this manuscript the authors present a hypothesis on how an evolutionary conserved Trp286/Trp434 clamp in the GH3 family facilitates processive hydrolysis of substrates. This hypothesis is backed up by data from biochemical assays, X-ray crystallography studies and computational chemistry. In the crystallographic approach a multitude of enzyme variants and substrate analogs have been thoroughly analyzed. The computational approach is broad, and the authors have selected suitable methods to investigate interactions between substrates, products, and enzyme variants. In this manuscript the authors have combined the powerful methods of structural biology and computational chemistry and have thus enabled analysis of molecular mechanisms at both at the macromolecular nano-scale to the dynamics of the electronic structures in the active site. This is an original and noteworthy contribution to the field.

Comments to the authors:

The amount of data presented in this manuscript is huge with several multipanel figures. I would suggest adding an overview figure where binding, hydrolysis and product release processes are illustrated in simpler terms to help the reader catch on.

A new Fig. 9 “Mechanistic view of the catalytic function of HvExol, a family GH3 enzyme” has been added to the manuscript and conferred in the Discussion section (page10, lines 402-403). The legend in Fig. 9 reads (page 29, lines 1185-1201):

“**Fig. 9** Mechanistic view of the catalytic function of HvExol, a family GH3 enzyme.

a HvExol uses the retaining catalytic mechanism, where nucleophile Asp285 (magenta) and acid/base catalyst Glu491 (green) play key roles in substrate hydrolysis¹⁻³. **b** Reaction kinetics indicates the formation of Michaelis complexes (E-S and E~S), and enzyme-product complexes before (E~P1~Glc) and following the first Glc egress (E~P1), and after hydrolysis of P1 (E~Glc)^{1,2}. S and P1 are trimeric and dimeric molecules, respectively. First-order rate constants are shown. Entrapped Glc product from E~Glc is displaced after a new substrate is attached. The hydrolytic cycle with a (1,3)-linked substrate is repeated until it is hydrolysed to Glc (indicated by vertical and horizontal lines). **c** Simplified mechanism of Glc displacement with a disaccharide²². After the disaccharide (empty blue and filled grey squares) bound in -1 and +1 subsites (step 1) is hydrolysed and the reducing end Glc (aglycon) diffuses away from +1 subsite, the non-reducing end Glc (cyan square) remains non-covalently trapped. After a new dimer binds (step 3) and advances to the catalytic site, Glc modifies its binding patterns and egresses (large arrow) via a transient lateral cavity (cylinder in dotted lines) formed near the Trp clamp (step 4)²². Curved arrows indicate dimer entries into the catalytic site. **d** Glc displacement paths in WT (left panels), and the W434A mutant (right panels) with Glc and the G3OG3 dimer, calculated through cMD and GPathFinder. Large arrows indicate Glc positions at initial (step 1) and after final (step 4 – following Glc departure from -1 subsites) stages of catalysis (this work).”

For the classical MD simulations there is no mention of the thermostat employed, this should be included. While the setup/equilibration for the cMD simulations described in lines 671-700 is accurately described, this is not the case for the cMD simulations described in lines 607-635.

Thank you for your suggestions. We have improved the description of cMD simulations of Michaelis complexes derived from thio-ligand complexes, in particular, the thermostat parameters used are now included (page 16, lines 643-648).

For the three different sections describing cMD simulations (starting at line 607, 621 and 671) different force fields have been applied (ff99SB, ff19SB with TIP3P instead of OPC water model, and ff99SB-ILDN). I would like the authors to comment on why different force field were applied in these simulations and if this has any consequences for the analysis of the results. For the GaMD simulations is not explicitly mentioned what force field that was applied.

The different AMBER force fields used are based on f99SB with modifications aimed at improving various aspects of this work. In principle, differences between these forcefields are not expected to have a significant impact on the findings and should not affect the conclusions of this work. Please notice that within the Methods paragraphs, the same forcefields have been used to explore a particular question and compare investigated systems. The usage of one or other forcefield used by different groups (with varying preferences) reflects on simulations that we run during various time intervals.

For GaMDs simulations, we used f99SB-ILDN, which is the same forcefield that we used for cMD simulations. The corresponding methods section has been modified and now reads (page 18, lines 733-734):
“For GaMD simulations⁵⁵, AMBER 18⁹⁸ was used with the same forcefield described in the previous paragraph.”

In all figures containing electron density maps (both in the main manuscript and supplementary information) there are no difference maps included. This should at least be explained or the figures should be updated so that the reader can get an visual impression of the quality of the modeling.

We have provided the information on how the difference electron density maps presented in all figures were generated. This specification was included in the Methods section (pages 15-16, lines 628-631). The new sentence reads:

“The $(2m|F_o|-D|F_c|)$ difference electron density maps were calculated using Phenix⁶⁷ with the mtz and PDB files after CCP4 REFMAC5 refinements. The maps were contoured at the 1.0 σ level. Ligands bound in active sites (Figs. 2-4; Supplementary Figs. 2-4) were visualised in PyMol Molecular Graphics System v2.3.3.”

Reference

67. Adams, P. D., Afonine, P. V., Bunkóczi, G., Chen, V. B., Davis, I. W. et al. PHENIX: a comprehensive Python-based system for macromolecular structure solution. *Acta Crystallogr.* **D66**, 213–221 (2010).

Also, in all the accompanying legends the term “ δ -level” is used while “ σ -level” is correct (change from delta-level to sigma-level).

Thank you for alerting us to errors (for us embarrassing) in the “delta-level” expression (“ δ -level”) in all structural figures. We apologise for our oversight. The correct term “1.0 σ level” has been introduced throughout the text and in all figures.

In supplementary Table 6 R-merge should be included for the highest/outer shell and the $CC_{1/2}$ parameter should be added for each structure.

Both values, that is R-merge (for the highest shell) and the $CC_{1/2}$ parameter values have been included in Supplementary Table 6, for all 25 crystal structures.

Considering the text below on data availability, I would prefer that the scripts were made publicly available at e.g. github and that a DOI was included. As a recent paper demonstrated – the reasonable request do rarely work out in practice.

« For analyses of geometrical parameters and MD simulations, we used in-house scripts, which are available upon reasonable request from the corresponding author. »

A script used to analyse the convergence of MD simulations and plot panels shown in Supplementary Figure 12a can be found at:

https://github.com/insilichem/utils/convergence_analysys_MD_pytraj.ipynb.

A script used to obtain, list, and plot interatomic distances along MD trajectories and plot individual panels shown in Supplementary Figure 12b can be found at:

https://github.com/insilichem/utils/interatomic_distances_MD_traj_graph.ipynb.

The scripts used to obtain and plot the interatomic distances and the Root-Mean-Square-Deviation values of the protein backbone, substrates, and residues along an MD trajectory shown in Supplementary Figure 4 can be found at:

<https://github.com/roviralab/utils/blob/main/analisi.tcl>

<https://github.com/roviralab/utils/blob/main/rmsd.py>

A script used to obtain cluster analysis of an MD trajectory shown in Fig. 5 can be found at:

<https://github.com/roviralab/utils/blob/main/cluster.ptraj>

A script used to plot the puckering free energy landscape shown in Fig. 5 and Supplementary Figure 8 can be found at:

<https://github.com/roviralab/utils/blob/main/mercator.py>

A script used to plot the conformation of the ring sugars as a function of θ and ϕ puckering coordinates shown in Supplemental Figure 7 can be found at:
https://github.com/roviralab/utis/blob/main/mercator_points.py.

Reviewer #3 (Remarks to the Author):

Reviewer report on "The evolutionary advantage of an aromatic clamp in plant family 3 glycoside exo-hydrolases" Luang et al have undertaken a very thorough study on the mechanism of processivity on a GH3 exo-hydrolase. The manuscript is well written and major conclusions are supported by well-designed experiments. Still, there is a serious lack in the presentation of the history in GH processive mechanism. Especially on the role of aromatic amino acids. I feel it is urgent that readers of this manuscript are informed that some of the major thoughts and findings presented in this work has already been shown by other research groups previously. This said, Luang et al provides structural and computational evidence for the role of aromatic amino residues in the processive mechanism that are new, and hence, deserves to have their work in Nature Communications. If the authors are able to include the findings from previous studies as support for their work and as a basis for what is already known, I will recommend that the work is published.

Comments to the authors:

Major points

- Line 78, Page 2. "Recently, we envisaged that β -D-glucan glucohydrolases, such as HvExoI, adopt so-called substrate-product assisted processive catalysis, where after an incoming substrate binds in a productive mode, the Glc product evokes the formation of a transient lateral cavity, which serves as a conduit for its egress to allow for the next catalytic round²¹." I have to say it is greatly disturbing to address this topic without citing papers such as Igarashi et al. Science 2011, Horn et al. PNAS 2006, and Zakariassen et al. J. Biol. Chem. 2009, which demonstrates that when the product dissociates, it is possible for a processive movement of the polysaccharide/oligosaccharide to occur.

Thank you for this suggestion. We are well aware of the extensive work done in the field of processive GH enzymes with tunnel-, groove-, ridge- or cleft-like catalytic sites. We have cited some of these key works in our previous work (Streltsov et al.; *Nat. Commun.* 2019; **10**, 2222; cf. page 12), and also in the first sentence of the Discussion section.

However, to our knowledge, the processive mechanism has not been described in GH enzymes with funnel-, crater- or pocket-shaped catalytic sites, before we published our first paper in *Nat. Commun.* 2019.

As suggested, and to put the information on processivity in GHs in a broader context, we have included new sentences and new references. Here, we cite additional key works in this field. This section reads as follows (pages 2-3, lines 84-92):

"To our knowledge, the type of processive catalysis (or non-dissociative sequential degradation) discovered in HvExoI²² has not been described in hydrolases with a closed pocket-shaped catalytic site. In doing so²², we defined processive catalysis (which we termed substrate-product assisted processivity) in broader terms than those used for the definitions of the tunnel-, groove-, ridge- or cleft-like active sites of GHs²³⁻²⁵. Considering that HvExoI operates on plant cell wall (1,3;1,4)- β -D-glucan polysaccharides, processive catalysis (as a form of efficient catalysis) in this enzyme is expected as rapid substrate hydrolysis is critical to embryo development during seed germination^{5,26}. Processive catalysis has been defined in (1,3)- β -D-glucan endohydrolases²⁷, (1,4)- β -D-glucan endohydrolases²⁸, cellobiohydrolases²³, and chitobiohydrolases^{24,25,29}."

References

22. Streltsov, V. A., Luang, S., Peisley, A., Varghese, J. N., Ketudat Cairns, J. R. et al. (2019) Discovery of processive catalysis by an exo-hydrolase with a pocket-shaped active site. *Nat. Commun.* **10**, 2222.
23. Igarashi, K. et al. Traffic jams reduce hydrolytic efficiency of cellulase on cellulose surface. *Science* **333**, 1279–1282 (2011).
24. Horn, S. J., Sikorski, P., Cederkvist, J. B., Vaaje-Kolstad, G., Sørlie, M., Synstad, B., Vriend, G., Vårum, K. M. & Eijsink, V. G. Costs and benefits of processivity in enzymatic degradation of recalcitrant polysaccharides. *Proc. Natl. Acad. Sci. USA* **103**, 18089–18094 (2006).
25. Zakariassen, H., Aam, B. B., Horn, S. J., Vårum, K. M., Sørlie, M. & Eijsink, V. G. H. Aromatic residues in the catalytic center of chitinase A from *Serratia marcescens* affect processivity, enzyme activity, and biomass converting efficiency. *J. Biol. Chem.* **284**, 10610–10617 (2009).
26. Hrmova, M. & Fincher, G. B. Structure-function relationships of β -D-glucan endo- and exohydrolases from higher plants. *Plant Mol. Biol.* **47**, 73–91 (2001).
27. Varghese, J. N. et al. Three-dimensional structures of two plant β -glucan endohydrolases with distinct substrate specificities. *Proc. Natl Acad. Sci. USA* **91**, 2785–2789 (1994).
28. Parsiegla, G., Reverbel, C., Tardif, C., Driguez, H. & Haser, R. Structures of mutants of cellulase Cel48F of *Clostridium cellulolyticum* in complex with long hemithiocello oligosaccharides give rise to a new view

- of the substrate pathway during processive action. *J. Mol. Biol.* **375**, 499–510 (2008).
29. Nakamura, A., Okazaki, K. I., Furuta, T., Sakurai, M. & Iino, R. Processive chitinase is Brownian monorail operated by fast catalysis after peeling rail from crystalline chitin. *Nat. Commun.* **9**, 3814 (2018).
 30. Sørensen, A., Lübeck, M., Lübeck, P. S. & Ahring, B. K. Fungal beta-glucosidases: a bottleneck in industrial use of lignocellulosic materials. *Biomolecules* **3**, 612–631 (2013).

- Line 84, Page 3. Again, it is well documented that aromatic amino acids are vital for the processive mechanism in GHs (Horn et al. PNAS 2006 and Jana et al. J. Phys. Chem. B. 2016).

We also have added a new sentence (page 3, lines 94-96) that reads:

“The work with these processive enzymes suggested the role of aromatic and polar amino acid residues to assist with stacking and hydrogen-bonding interactions in polysaccharide sliding motions through the catalytic sites^{22-25,27-29,31}.”

Reference

31. Hamre, A. G., Jana, S., Reppert, N. K., Payne, C. M. & Sørlie, M. Processivity, substrate positioning, and binding: the role of polar residues in a family 18 glycoside hydrolase. *Biochemistry* **54**, 7292–306 (2015).

- Line 116, Page 3. "Trp286/Tyr286 and Gln434/Tyr434/Trp434 in subclade 2, and Gly434/His434/Lys434/Trp434 in subclade 3 (Fig. 1c), which may impact substrate binding and processivity in these enzymes." The role of aromatic amino acids in the processive mechanism of GHs have clearly been demonstrated in Horn et al. PNAS 2006 and Jana et al. J. Phys. Chem. B. 2016. Moreover, the role of polar amino acids (non-aromatic amino acids) has previously been described by Hamre et al. Biochemistry 2015. The authors are not the first to propose this.

The response to this suggestion is described in the response to previous suggestions. The specific reference (Hamre et al. Biochemistry 2015) was cited, as described above.

Minor points

- Line 50, Page 2. Would it be beneficial to introduce the Figure of HvExol already here as it is detailed important information of the structure and function?

Thank you for this excellent suggestion to improve the flow of information. We have added the reference to Figs. 1a, 1b, as suggested (page 2, line 51).

- Line 56, page 2. " while the reducing Glc moiety at the +1 subsite is constrained weakly,". I understand that this is citations of previous works, but stacking of sugar moieties with aromatic amino acids are not weakly. These can be in the range of 2 to 5 kcal/mol (Zolotnisky et al. PNAS 2004, Jana et al. J. Phys. Chem. B. 2016). We have replaced the word “weakly” with “through stacking interactions” (page 2, line 58). This sentence reads: “The capacity of HvExol to hydrolyse a panel of β -D-glucosides was rationalised from the observation that non-reducing Glc at the -1 subsite is locked into a fixed position through a network of hydrogen bonds (H-bonds), while the reducing Glc moiety at the +1 subsite is constrained through stacking interactions, and thus the flexibility of attached substrates and their projection is unencumbered allowing HvExol to accommodate isomeric (1,2)-, (1,3)-, (1,4)- and (1,6)-linked β -D-glucosides^{2,3}.”

Thank you also for the suggestion regarding the reference Zolotnisky et al. (PNAS 2004), which will make the statement quantitative. We have added a new sentence (page 2, lines 60-61):

“It was estimated that the strength of binding through aromatic residues could be in the range of 2-5 kcal/mol⁷.”

Reference

7. Zolotnisky, G., Cogan, U., Adir, N., Solomon, V., Shoham, G. & Shoham, Y. Mapping glycoside hydrolase substrate subsites by isothermal titration calorimetry. *Proc. Natl. Acad. Sci. USA* **101**, 11275–11280 (2004).

- Line 153, Page 4. "These data signal that although the mutations of Trp434 exerted a stronger inhibitory trend compared to Trp286, both Trp residues are required for binding isomeric β -d-glucosides into the productive binding modes required for high catalytic efficiency." Is the word "inhibitory" appropriate as this is normally associated with enzyme inhibition?

Thank you, we agree. This term was inappropriate in the context and was replaced with “affected the catalytic properties of HvExol more substantially” (page 5, lines 170-172).

- Line 158, Page 4. "to quantify dissociation constants (K_i)" K_d is the dissociation constant, K_i is the inhibition constant.

Thank you for the note. We have corrected the term and used "apparent inhibition constant" (page 5, line 176).

- Line 166, Page 4 and 5. "while the most effective thio-inhibitor was G6SG-OMe, except that it did not bind to W434A and showed around 6- and 10-fold lower K_i constant values with W434H and W434F, respectively (Supplementary Table 4)." Please address with a short line in the text why this is so.

We have added a summarising sentence to this paragraph (page 5, lines 185-187). It reads:

"These data pointed out that the replacements of Trp434 with Ala, His, or Phe residues in the +1 subsite, disturbed the optimal binding of thio-analogues due to the removal or weakening stacking interactions crucial for correct positioning of inhibitors in the active sites."

- Line 185-186, page 5. "4NP-G3SG bound tightly to WT HvExol with the respective K_i constants of 382×10^{-6} M and 377×10^{-6} M for recombinant and native enzymes." Do you have three significant number in your experiments? This is not likely. Unless you report standard deviations, it should be 0.4×10^{-9} M. This goes as well to (line 176) K_i of $3,270 \times 10^{-6}$ M and many others. Check throughout the manuscript.

Thank you for this suggestion, and we agree. We have rounded the K_i constant values to one significant digit in Supplementary Table 4. We have re-checked and introduced minor corrections in the numerical data.

We have also re-checked the entire text and replaced all K_i constants values with rounded values. These instances are: on page 5, lines 195 and 205; on page 6, lines 217-218, and 234.

REVIEWERS' COMMENTS

Reviewer #2 (Remarks to the Author):

Thank you for considering my suggestions. I think all my previous comments has been addressed in a constructive manner.